# Reviews and syntheses: A framework to observe, understand, and project ecosystem response to environmental change in the East Antarctic Southern Ocean

Julian Gutt[1], Stefanie Arndt[1], David Keith Alan Barnes[2], Horst Bornemann[1], Thomas Brey[1,3], Olaf Eisen[1,4], Hauke Flores[1], Huw Griffiths[2], Christian Haas[1], Stefan Hain[1], Tore Hattermann[5], Christoph Held[1], Mario Hoppema[1], Enrique Isla[6], Markus Janout[1], Céline Le Bohec[7,8], Heike Link[9], Felix Christopher Mark[1], Sebastien Moreau[5], Scarlett Trimborn[1], Ilse van Opzeeland[1,3], Hans-Otto Pörtner[1], Fokje Schaafsma[10], Katharina Teschke[1,3], Sandra Tippenhauer[1], Anton Van de Putte[11,12], Mia Wege[13], Daniel Zitterbart[14,15], Dieter Piepenburg[1,3,16]

[1]Alfred Wegener Institute, Helmholtz Centre for Polar and Marine Research, 27570 Bremerhaven, Germany

[2]British Antarctic Survey, Cambridge, CB3 OET, UK

[3]Helmholtz Institute for Functional Marine Biodiversity, Ammerländer Heerstraße 231, 26129 Oldenburg, Germany

[4]Geosciences, University of Bremen, 28359 Bremen, Germany

[5]Norwegian Polar Institute, Hjalmar Johansens gate 14, 9007, Tromsø, Norway

[6]Institute of Marine Sciences-CSIC, Barcelona, 08003, Spain

[7]Université de Strasbourg, CNRS, IPHC UMR 7178, F-67000, Strasbourg, France

[8]Centre Scientifique de Monaco, Département de Biologie Polaire, MC 98000, Monaco City, Monaco

[9]Department Maritime Systems, University of Rostock, 18059 Kiel, Germany

[10]Wageningen Marine Research, Ankerpark 27, 17871 AG Den Helder, The Netherlands

[11]Royal Belgian Institute for Natural Sciences, Brussels, Belgium

[12]Université Libre de Bruxelles, Brussels, Belgium

[13]Mammal Research Institute, Department of Zoology & Entomology, University of Pretoria, Hatfield Pretoria, 0002, South Africa

[14]Applied Ocean Physics and Engineering Department, Woods Hole Oceanographic Institution, Woods Hole, MA, 02543, USA

[15]Friedrich-Alexander-Universität Erlangen-Nürnberg, 91054, Erlangen, Germany

[16]Institute for Ecosystem Research, University of Kiel, 24118 Kiel, Germany

*Correspondence to*: Julian Gutt (julian.gutt@awi.de)

**Abstract.** Systematic long-term studies on ecosystem dynamics are largely lacking from the East Antarctic Southern Ocean, although it is well recognized that they are indispensable to identify the ecological impacts and

risks of environmental change. Here, we present a framework for establishing a long-term cross-disciplinary study
on decadal time scales. We argue that the eastern Weddell Sea and the adjacent sea to the east, off Dronning Maud
Land, is a particularly well-suited area for such a study, since it is based on findings from previous expeditions to
this region. Moreover, since climate and environmental change have so far been comparatively muted in this area,
as in the Eastern Antarctic in general, a systematic long-term study of its environmental and ecological state can
provide a baseline of the current situation, which will be important for an assessment of future changes from their
very onset, with consistent and comparable time series data underpinning and testing models and their projections.
By establishing an "Integrated East Antarctic Marine Research" (IEAMaR) observatory, long-term changes in
ocean dynamics, geochemistry, biodiversity and ecosystem functions and services will be systematically explored
and mapped through regular autonomous and ship-based synoptic surveys. An associated long-term ecological
research (LTER) programme, including experimental and modelling work, will allow for studying climate-driven
ecosystem changes and interactions with impacts arising from other anthropogenic activities. This integrative
approach will provide a level of long-term data availability and ecosystem understanding that are imperative to
determine, understand, and project the consequences of climate change and support a sound science-informed
management of future conservation efforts in the Southern Ocean.
**1 Introduction**
**1.1 Background**
Life in the Southern Ocean (SO) significantly contributes to global marine biodiversity and ecosystem services
(Kennicutt et al., 2019; Steiner et al., 2021) and is, thus, of substantial importance for the global climate, biosphere
and human wellbeing (Grant et al., 2013; Cavanagh et al., 2021). However, there is growing evidence that the
Southern Ocean, like polar regions in general, is particularly sensitive to the impacts and risks of environmental
change, as highlighted, e.g., in the "6[th] Assessment Report of the Intergovernmental Panel on Climate Change
(IPCC)" (IPCC, 2022) and, specifically, in the "IPCC Special Report on the Ocean and Cryosphere in a Changing
Climate" (Meredith et al., 2019) as well as the "Antarctic Climate Change and the Environment" report (ACCE)
of the "Scientific Committee on Antarctic Research" (SCAR) (Turner et al., 2014). In a joint report the IPCC and
the "Intergovernmental Science-Policy Platform on Biodiversity and Ecosystem Services" (IPBES) assessed the
impact of climate change on global biodiversity in relation to land and ocean use and predicted that the proportion
of climate change related biodiversity impacts will increase in the next decades (Smith et al., 2022). Due to the
vast, remote, and harsh nature of the environment in the Antarctic region, any comprehensive observation system
requires international collaboration to establish and provide access to infrastructure and data.
Despite increased scientific interest and efforts, the scientific community has recognized major knowledge gaps
regarding the vulnerability of SO biotas to anthropogenic impacts and risks, especially those driven by climate
change (Flores et al., 2012; Vernet et al., 2019; Gutt et al., 2021). Such information is urgently needed to develop
high-confidence projections of future ecosystem changes (Kennicutt et al., 2014; Pörtner et al., 2021) and to be
able to support targeted action to mitigate or adapt to such changes, as also recently requested in the Southern
Ocean Action Plan in support of the UN Decade of Ocean Science for Sustainable Development (Janssen et al.,
2022). SCAR also supports the "Southern Ocean Observation System" (SOOS) initiative, the "SCAR Antarctic
Biodiversity Portal" (https://www.biodiversity.aq, last access: 23 August 2022) and has recently launched the
scientific research programme "Integrated Science to Inform Antarctic and Southern Ocean Conservation" (Ant-
ICON). Together, these actual and previous research efforts provide the best possible international scientific basis
for climate-change detection and attribution, as well as for decision-making with respect to nature conservation
in the Antarctic by the "Committee for the Conservation of Antarctic Marine Living Resources" (CCAMLR).
Long-term observatories have already been established in the Arctic and Antarctic, providing valuable information
on mainly climate-driven shifts in and drivers of biodiversity and biological processes. However, only a small
number of them are located in the East Antarctic, the larger area of interest of the concept presented here.
Moreover, they are all thematically rather narrow and mono-disciplinary in scope, and they were carried out
independently from each other (see 4.1).

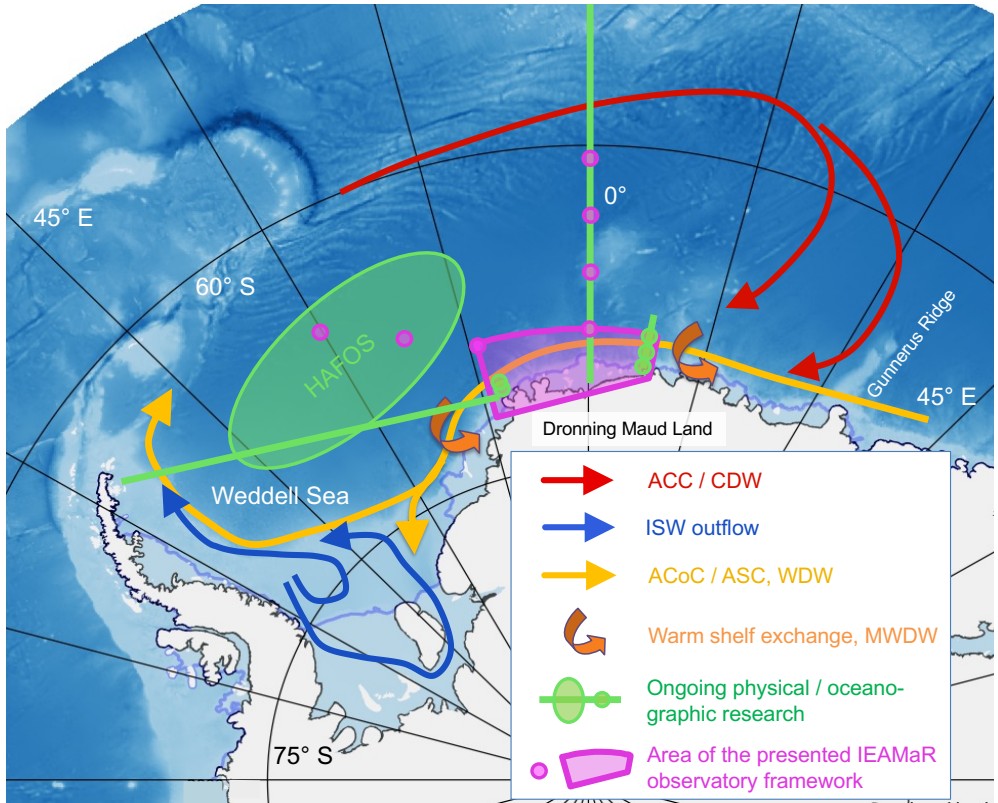


**Figure 1:** Possible location of the concept for an "Integrated East Antarctic Marine Research" (IEAMaR)
observatory within the East Antarctic Southern Ocean. Arrows indicate large-scale advective water mass
pathways. Deep water entering the Weddell gyre from the ACC joins the southern limb of the gyre, of which the
ACoC is also a part. After leaving the IEAMaR region, the water flow continues along the slope and shelves.
Interaction with the broad shelves in the south leads to ISW, a predecessor of Antarctic Bottom Water. Small
green circles indicate sites of ongoing mooring programs. ACC/CDW: Antarctic Circumpolar
Current/Circumpolar Deep Water, ISW: Ice Shelf Water, ACoC/ASC: Antarctic Coastal Current/Antarctic
Slope Current, MWDW: Modified Warm Deep Water. Only approximate location for the HAFOS (Hybrid
Antarctic Float Observing System) area indicated. Design: Tore Hattermann.

## 1.2 Knowledge gaps

For a comprehensive assessment of climate-change impacts and evidence-based action recommendations, the current scientific knowledge in terms of a quantification of physical-chemical ecosystem drivers, an understanding of ecosystem processes and of temporal shifts of biodiversity as well as its spatial heterogeneity, is insufficient for a number of reasons. Firstly, the impacts of climate change and other anthropogenic activities are not uniform, in space, time and across organisms (Rogers et al., 2020). Secondly, a whole-ecosystem response to external forcing and disturbances is generally difficult to assess in "end-to-end" observations and simulations (i.e., from primary production and its drivers to apex predators) (Walther et al., 2002) given that environmental stress cascading through the ecosystem is non-linear. Thirdly, advanced tools were not available and important background information did not exist in the past. Fourthly, some modern research strategies and their implementation that address the following gaps of knowledge and knowledge transfer have not yet gained sufficient acceptance:

(1) Synoptic surveys generating long-term and year-round data series and allowing an assessment of complex climate-induced changes (vs natural variability) are lacking (IPCC, 2022).

(2) Although concepts for standardized protocols, operating procedures and data integration do exist (see e.g., Miller et al., 2015; Piazza et al., 2019; Van de Putte et al., 2021), they have not been frequently and consequently implemented. They have to be urgently applied to acquire large-scale and long-term comparable biogeographic data.

(3) An integration of multi-disciplinary data derived from experiments as well as digital and genomic analyses in coupled atmosphere-ocean-cryosphere-biosphere models is still in its infancy. Such models, however, can provide deeper insights in ecosystem functioning and carbon sequestration under specific climate change and protection scenarios (Gutt et al., 2018).

(4) The impacts of multiple and cascading stressors, e.g., how climate change amplifies fishing impacts or combined effects of sea-ice shrinking, ocean warming and ocean acidification, are so far only poorly studied (Kennicutt et al., 2014; Gutt et al., 2015). Such knowledge is needed, however, for a sound understanding of whole-ecosystem functioning and to recognize synergistic effects.

(5) The awareness of the contributions of SO biotas to global ecosystem services is still insufficient among stakeholders and decision makers to assess their value in a global context.

## 1.3 Objectives

To address these knowledge gaps, we

(1) emphasize the urgent need of cross-disciplinary research and synoptic surveys related to environmental changes in sea ice and the water column, at the sea-floor and the underside of floating ice shelves, developing the framework for a long-term research observatory in the Eastern SO,

(2) lay out a conceptual framework for upcoming work, time and cost plans, hereafter called "Integrated East Antarctic Marine Research" (IEAMaR) observatory,

(3) justify its placement in the eastern Weddell Sea and western part of the sea off Dronning Maud Land (Fig. 1), and

(4) describe three scientific themes addressed by the long-term observations and complementary scientific studies
131       to be performed at the observatory.


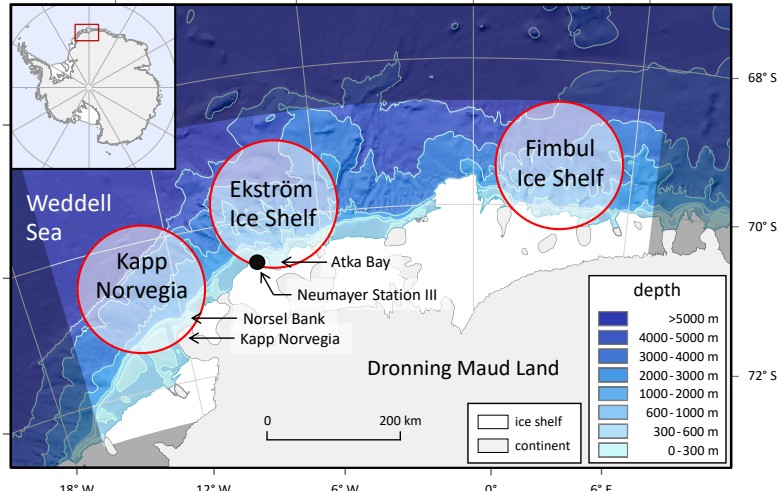


**Figure 2:** (a) Geographic position of the components of a possible long-term "Integrated East Antarctic Marine
Research" (IEAMaR) observatory. The three circles represent three possible sub-areas, off the Fimbul and
Ekström ice shelves, respectively, as well as off Kapp Norvegia. The highlighted area shows the wider IEAMaR
region, where large-scale data from methods like remote sensing and bathymetry are important for most other
specific measurements. Bathymetric colour codes refer to the highlighted cutout; Bathymetry south of 60°S:
Arndt et al. (2013); continent and ice shelf: https://www.npolar.no/quantarctica/ (last access: 23 August 2022).
Design: Rebecca Konijnenberg, Hendrik Pehlke, and Julian Gutt.

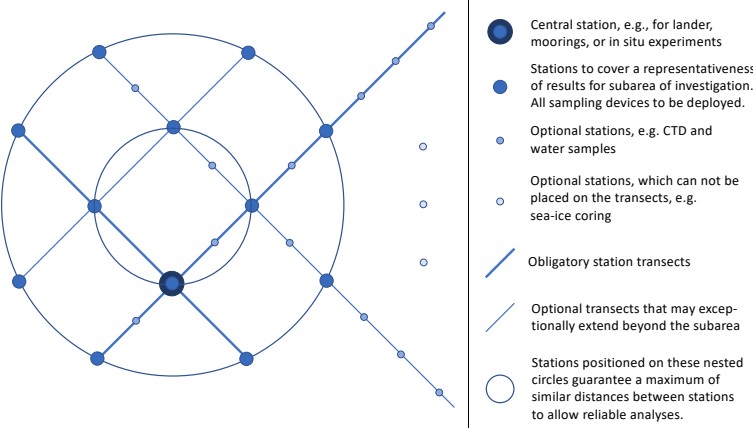


(b) Schematic design of the positioning of stations/transects within each of the three IEAMaR sub-areas to be
sampled.

These objectives can be best addressed by establishing a collaborative IEAMaR long-term observatory in the eastern Weddell Sea and adjacent western part of the sea off Dronning Maud Land (DML), approx. south of 69°S between 16°W and 6°E (Figs. 1 and 2). Regular observational work should be conducted over a period of decades, and the observatory should provide a platform for an integrated cross-disciplinary "Long-Term Ecological Research" (LTER) programme to generate reliable fact-based evidence for changes in SO ecosystems, and the role of anthropogenic causes, especially climate change driving these changes. The long-term observatory represents the location and logistic infrastructure of the intended observational work, while LTER refers to the scientific studies to be carried out there. Such a rigorous cross-disciplinary "biodiversity exploratory" approach (combination of observations and first-principle process studies) has been shown to be particularly suited to identify, describe, gauge, understand and project the processes driving temporal ecological changes and spatial habitat-turnover in representative regions and habitats (Fischer et al., 2010). In addition, a separation of intrinsic oscillations in physical, geochemical and biological processes from extrinsic trends is necessary to attribute observed variability to climate change, inform stakeholders and educate future generations of polar researchers (Fig. 3). A detailed system understanding is to be enhanced through downscaling approaches, studying detailed key ecosystem functions (production, export, and biogeochemical cycles) and species-specific processes, interactions and adaptations (species distribution and range shifts, behavioural and phenological adaptations, physiological acclimation and genetic mutations). Upscaling results from specific sites will improve our knowledge on regional biodiversity including temporal shifts and allow to model coupled physical-biological projections, which is important for the large-scale assessments of the IPCC, IPBES and the "World Ocean Assessment" of the UN, as well as scientific advisory bodies, such as SCAR, CCAMLR and the "Committee for Environmental Protection" (CEP), the two latter being part of the Antarctic Treaty System. Fishing in the wider Weddell Sea region is currently limited to exploratory fishing of Antarctic toothfish (*Dissostichus mawsoni*) off Dronning Maud Land. Although the intention was expressed some years ago to also conduct exploratory fisheries for Antarctic krill in this region, no krill is currently fished there. The IEAMaR area would overlap considerably with the proposed "Weddell Sea Marine Protected Area" (WSMPA) and provide an important key hub for the required research and monitoring to be carried out according to a WSMPA management plan.

172

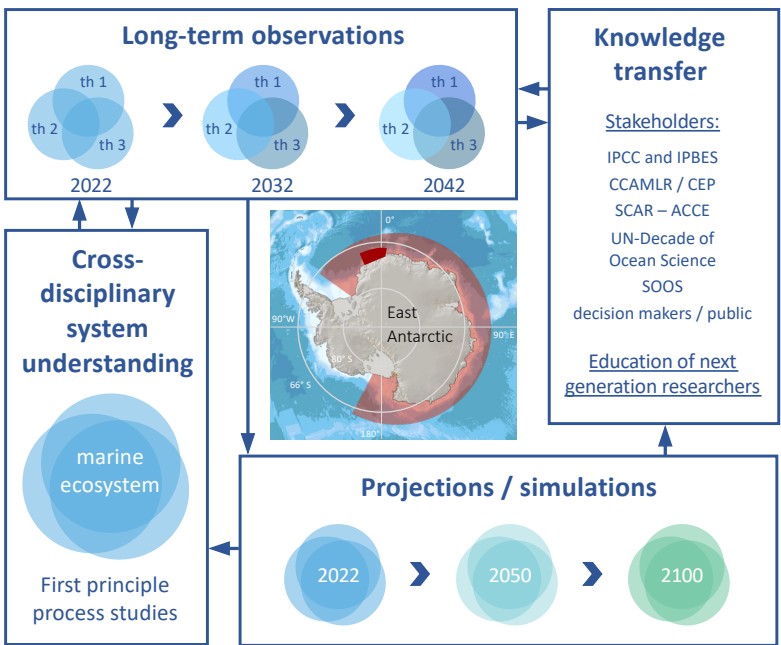

173

**Figure 3:** Relationships between approaches, objectives and potential stakeholders of presented "Integrated East Antarctic Marine Research" (IEAMaR) observatory. The dark red rectangle in the map in the centre indicates where the observatory can be placed within the East Antarctic coastal Southern Ocean (light red, transparent). IPCC: Intergovernmental Panel on Climate Change; IPBES: Intergovernmental Science-Policy Platform on Biodiversity and Ecosystem Services; CCAMLR: Committee for the Conservation of Antarctic Marine Living Resources; CEP: Committee for Environmental Protection; SCAR: Scientific Committee on Antarctic Research; ACCE: Antarctic Climate Change and the Environment; SOOS: Southern Ocean Observing System; th: ecological research theme.

182

We argue that the IEAMaR observatory is urgently needed because the recent relative environmental stability of East Antarctica provides a reliable baseline for climate-related ecosystem parameters that can be used to underpin and calibrate projected biological changes caused by climatic and non-climatic drivers.

## 2 Overarching concept

### 2.1 Geographical and environmental justification

For the following reasons, the IEAMaR region is particularly suited for performing LTER to detect and understand ecological changes and predict the future developments of the coupled atmosphere-cryosphere-ocean-biosphere system in the East Antarctic SO on a decadal time scale (see also Lowther et al., 2022):

(1) The region is characterized by high-latitude conditions of which most are typical for the East Antarctic, including a coast shaped by a glaciated land mass and ice shelves, bounded by ice rises, rumples, and small islands stabilizing the ice shelves (Matsuoka et al., 2015) but also leading to more complex circulation underneath them (e.g. Smith et al., 2020), frequent calving, transiting and grounding of icebergs, specific water masses, and high inter-annual as well as intra-annual variation in the seasonal sea-ice cover and primary production. For examples of some most important environmental drivers of the marine ecosystem in the area under consideration see Fig. 1 (currents), Fig. 2 (bathymetry), and Fig. 4 (sea ice, sea surface temperature, and chlorophyll-a).

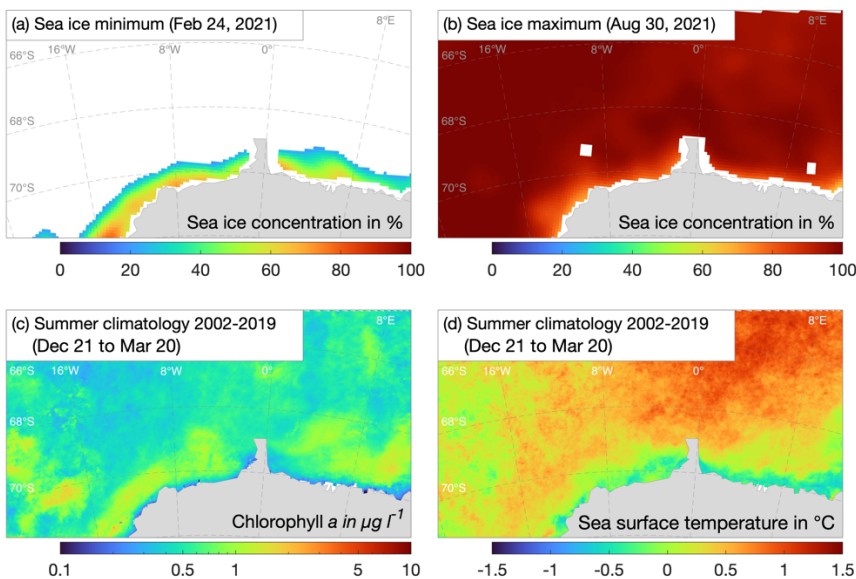

**Figure 4:** Sea ice concentration for the minimum and maximum sea ice extent in 2021, and the summer climatology, i.e. December 21 to March 20, for the time period 2002-2019 for the sea surface chlorophyll a concentration and temperature in the region of the "Integrated East Antarctic Marine Research" framework. Sea ice concentration data were obtained from EUMETSAT Ocean and Sea Ice Satellite Application Facility (OSI SAF, Lavergne et al., 2019). Sea surface temperature and chlorophyll a concentration were obtained from the ocean color data distribution site: http://oceandata.sci.gsfc.nasa.gov/ (last access: 23 August 2022).

(2) Large-scale oceanographic features potentially exposed to climate change impact this region (Fig. 1): The Weddell Gyre branches off the eastward flowing Antarctic Circumpolar Current (ACC; van Heuven et al., 2011) and converges between Gunnerus Ridge (30° East) and the Ekström Ice Shelf (8° West) with the westward flowing Antarctic Coastal Current (ACoC) near the coast and ice shelf fronts and the Antarctic Slope Current (ASC) along the continental slope facilitating zonal connectivity and shaping the coastal environment. An overturning

circulation (Jullion et al., 2014) with strong links to the carbon cycle (MacGilchrist et al., 2019) is associated with this circulation that is driven by winds and modulated buoyancy fluxes due to sea-ice melting, freezing and ice-shelf-ocean interactions.

(3) Similar to most regions in the East Antarctic SO (east of 20°W), climate-driven changes in the wider IEAMaR region are currently insignificant or less pronounced than further north and along the Antarctic Peninsula (Turner and Comiso, 2017). However, there is evidence for some initial changes in the East Antarctic SO (Eayrs et al., 2021). Profound and widespread climate change (IPCC, 2021), with severe ecological impacts (IPCC, 2022), is projected under all climate scenarios. Both warming (Kusahara and Hasumi, 2013) and freshening (de Lavergne et al., 2014) of coastal waters are projected in the IEAMaR region, with interactions and feedbacks that may further enhance access of Warm Deep Water into the eastern (Hattermann, 2018) and southern WS (Hellmer et al., 2012; Daae et al., 2020) and melting of the Filchner-Ronne Ice Shelf (Timmermann et al., 2017) with unpredictable consequences for the marine ecosystem. Expected and already observed changes in the central and western Weddell Gyre include ocean acidification and a freshening of surface and deep waters (Jullion et al., 2013).

(4) Sea ice, which shapes the entire marine ecosystem, has slightly increased in extent in the East Antarctic SO over the past decades, albeit with strong interannual variations. The unprecedented springtime retreats in 2016 and 2021/22 (Turner et al., 2017 and 2022) and generally lower summer extent since 2016 and 2021/2022 (compared to the 1981/2010 long-term mean) may indicate the onset of a circum-Antarctic decline of sea-ice extent (Rackow et al., 2022). Increased upwelling, probably associated with the Southern Annular Mode, is the most reasonable explanation for changes in nutrient concentrations in the upper water column in the Weddell Gyre since the 1990's (Hoppema et al., 2015). This may explain an increase in sea surface phytoplankton biomass between 1997 and 2020 (Pinkerton et al., 2021).

(5) Previous studies have shown that the IEAMaR region houses a variety of habitats, which are representative of East Antarctic seas: neritic and oceanic pelagic, benthic and sympagic communities, overdeepened basins (innershelf depressions), flat shelf areas, a glaciated coast, a coastline formed by floating ice shelves with an almost unstudied underside and marine seabed and ice rises underneath, inlets in the ice shelves, iceberg grounding zones, fast-ice, pack-ice, and unusually shallow banks.

(6) For an East Antarctic region, the suggested area is comparatively well explored, as it has been subject to regular marine research expeditions for over 40 years, such as, e.g., the "European Polarstern Study" initiative (EPOS; Hempel 1993), the SCAR program "Ecology of the Antarctic Sea Ice Zone" (EASIZ; Arntz and Clarke, 2002; Clarke et al., 2006) and German-led national programs and expeditions such as the "Hybrid Antarctic Float Observing System" (HAFOS) and "Continental Shelf Multidisciplinary Flux Study" (COSMUS) as well as the Norwegian expedition "Mind the gap: Bridging knowledge and decision-making across sectoral silos and levels of governance in ecosystem based management" (ECOgaps). Some of the ongoing studies are summarized by de Steur et al. (2019). The data gained during these investigations will provide a valuable knowledge base, which the long-term IEAMaR programme can build on. These studies have mostly been conducted during the austral summer, while only a few targeted multi-year dynamics, such as surveys in the Larsen A/B ice shelf areas between 2007 and 2011, the "Benthic Disturbance Experiment" (BENDEX) starting in 2003 or the "Lazarev Sea Krill Study" (LAKRIS) expedition (2004-2008). Data from the IEAMaR study area (Fig. 2) were compiled as a basis

for the proposal for a "Weddell Sea Marine Protected Area" (WSMPA; Teschke et al., 2020a; 2020b). Basic circumpolar biogeographic and biodiversity knowledge were published in the biogeographic atlas of the "Census of Antarctic Marine Life" (De Broyer et al., 2014; Van de Putte et al., 2021).

(7) The East Antarctic SO is relatively pristine with respect to noise, pollution, fisheries and tourism.

**2.2 Methodological approach: Observations, experiments, and models**

To address the presented objectives, a combination of observational work, experimental studies, data-integration and modelling, conducted at various spatial and temporal scales, will be applied.

The IEAMaR long-term observatory shall consist of a sensitive "change-detection" array at a number of sites/stations distributed along ecologically important gradients (Distributed Biological Observatory approach; Moore and Grebmeier, 2018) within up to three sub-areas (Fig 2). Standardized "ecosystem Essential Ocean Variables", eEOV as described by Constable et al. (2016) and "Essential Biodiversity Variables" (EBV; Pereira et al., 2013), will be observed and compared at regular time intervals, e.g., species abundance and composition, reproduction and growth, as well as fishery and pollution pressure.

Such surveys are the core of the long-term observatory concept, integrating data from various complementary approaches and sources (autonomous long-term in-situ monitoring, regular ship-based sampling, satellite-based remote sensing). Shipboard and autonomous data collections are suggested to take place at water depths ranging from the coastal shelf, including unusually shallow sites like the Norsel Bank off Kapp Norvegia (approx. 60 m), the slope between approx. 450 and 3000 m depth, influenced by the ACoC, to the deep sea, influenced by the Weddell Gyre. The concept envisages up to three sub-areas, since these should cover the different physical-chemical prerequisites and the biological heterogeneity within the wider study area, being partly representative for the East Antarctic SO (for overview see section 2.1, for details see information provided by the three scientific themes in sections 3). Their final number, actual position and size of the sub-areas (Fig. 2) need to be defined after careful revision of environmental settings, spatial ecological heterogeneity, and detailed requirements of the LTER concept. Thereby, environmental (physical and chemical) and biological information can be gained at a range of spatial and temporal resolutions, to assess changes at scales of years to decades through regular ship-based surveys. These combined measurements can resolve the timing of interlinked, strongly seasonal processes and episodic extreme events by complementing ship-based snap-shot measurements with year-round high-frequency (hourly to weekly) observations of selected variables obtained through autonomous installations such as moorings, landers and satellites (e.g., physical measurements, environmental DNA analyses and Chlorophyll-a). Moreover, historical transects, such as the Prime Meridian, should be extended (Fig. 1).

Along the IEAMaR transects, shipboard work should be carried out at regular, if possible, yearly, intervals, using standardized sampling protocols during cruises of ice-going research vessels, such as RV *Polarstern* or others. Focus of observational work should be on the systematic sampling of four types of "Essential Variables" (EVs) implemented by the scientific community: essential climate (ECV), ocean (EOV), biodiversity (EBV) and Ecosystem (EEV) variables (Van de Putte et al., 2021). This comprehensive approach would require the utilisation of a wide range of sampling methods, including casts of Conductivity Temperature Depth probes (CTD), pelagic and benthic catches and video observations (for more details see section 3 below), at fixed stations arranged in specific predefined patterns allowing for replicated sampling at different spatial scales (Fig. 2b for a possible

design of the spatial arrangement of stations and transects), which is necessary to ensure temporal and spatial comparability as well as representativeness for a larger area. The shipboard work would complement the higher-resolution observations performed by autonomous platforms at selected core stations, to be placed at the centres of the long-term observation transects or at existing long-term observation transects (Weddell Sea/Kapp Norvegia, Prime Meridian), to allow for the technical maintenance of the autonomous platforms and contribute to the ground-truthing of remote-sensing and modelling studies. The platforms can include various systems, such as moorings, profilers, saildrones, sea-ice buoys, gliders, benthic landers, underwater fish observatories, and time-lapse cameras, with the potential to grow into a network of autonomous observation devices. In addition, experimental work on specific objectives can be performed during the cruises, and top predators (seals and penguins) can be equipped with CTD-satellite trackers and biologging devices.

A nearby land- or ice shelf-based research station can be used for technical supply of underwater equipment used during shipboard work, deployment and retrieval of long-range autonomous underwater vehicles and maintenance of autonomous observatories as well as coastal glaciologic studies. The use of lab facilities of the research station for experiments and routine collection of local biotas has advantages over shipboard work. From the base, field work including the sampling can be conducted by means of mobile sledge-based container systems, e.g., a diving hut, an aquarium container. The ice-shelf associated fauna can be monitored by sensors and cameras attached to moorings and frozen in ice shelf boreholes. Acquired data can be sent by cable to a recording station on the surface of the ice shelf where the data storage and energy supply is located. The Neumayer Station III would be well suited to serve as such a base, as well as for managing the observatory in the Ekström Ice Shelf sub-area.

As a suitable prerequisite, the bathymetry in this area is very well known, from swath sonar in the open ocean, also underneath the ice shelf from active seismic surveys (Oetting et al., 2022). Remote sensing data at large spatial and small temporal scales can be easily acquired from routine satellite observations of the IEAMaR area for a number of ecologically relevant variables, such as sea-ice concentrations, ice types, drifts and deformation, sea-ice thickness, polynya activity, and primary productivity. The systematic collection of satellite imagery can also be used to monitor penguin and seal abundance to understand how environmental stochasticity influences the distributions and numerical abundance of sentinel species (e.g., LaRue et al., 2022; see also section 3.2.3). The cross-disciplinary studies at the IEAMaR observatory will benefit from meteorological routine measurements and glaciological data obtained from observations and from satellite ice shelf altimetry, including basal melt rates at the Neumayer Station III.

There are challenges regarding the implementation of the LTER programme. The research equipment to be used is mostly already available, indeed, but some devices need further technical development and targeted modification, e.g., regarding autonomous long-term recording of biological data with imaging methods or the application of genomic technologies (Brandt et al., 2016). Moreover, an adequate design of the replicate sampling (Fig. 2b) is of crucial importance for providing representative data that can be used for spatial upscaling and the intended spatial and temporal comparisons (Jurasinski and Beierkuhnlein, 2006). Existing approaches must be customized for the specific conditions of the study area and the type of data acquired (e.g., seabed imaging along transects). Last but not least, the extreme high-latitude Antarctic conditions have to be taken into account, since there is quite a high likelihood of time-series data losses to occur because sampling stations may not be accessible

at regular intervals due to changing sea-ice conditions, and autonomous platforms (moorings or landers) are lost
due to collisions with drifting icebergs.
**2.3 Data management**
For the cross-disciplinary approach, with various data to be integrated into an aggregated question-driven data
product, an appropriate data management system is essential. It should address the specific properties of the
Southern Ocean, be compatible in a global context and make use of existing data platforms and standards. Its data,
algorithms, and tools should rigorously apply the principles of FAIR (Findable, Accessible, Interoperable, and
Reusable; Wilkinson et al., 2016) and TRUST (Transparency, Responsibility, User Community, and
Sustainability and Technology; Lin et al. 2020; Van de Putte et al., 2021). It should be centred around Essential
Variables (EOVs, EBVs, ECVs, EEVs, and eEOVs), linked to the International Polar Year (IPY data vision) and,
more recently, follow the principles put forward by the Polar Data policies. Biodiversity data should follow the
Darwin Core standard developed by the Biodiversity Information Standards consortium, which is also used by the
Ocean Biodiversity Information System and Global Biodiversity Information Facility (Beja et al., 2021). This
biodiversity standard originally focused on information on preserved specimens but is now capable of providing
comprehensive metadata and links to other forms of data such as image repositories and molecular data.
**3 Three long-term ecological research (LTER) themes**
**3.1 The physical-chemical environment: ecosystem drivers**
**3.1.1 Background**
At the narrow continental shelf along the DML coast, the quasi-circumpolar westward flowing water masses
dominated by the ACoC and the ASC converge in the IEAMaR area into a coherent boundary current system
(Nunez-Riboni and Fahrbach, 2009; Le Paih et al., 2020). While the interior basin is a large contiguous region of
upwelling, the wind-driven downwelling over the eastern continental shelf maintains a pronounced slope front
(Heywood et al., 1998) that protects the glaciated coast from Warm Deep Water (WDW), a regional derivative of
Circumpolar Deep Water (CDW) that is brought southward in the eastern branches of the Weddell Gyre. Coastal
waters are part of the "fresh shelf" regime (Thompson et al., 2018), where interactions with the adjacent ice shelves
are controlled by a seasonal interplay between wind-driven downwelling of solar-heated surface water (Zhou et
al., 2014) and cross-front exchanges of modified WDW (mWDW) at depth (Nøst et al., 2011; Hattermann et al.,
353 2014).

Direct observations and estimates of basal melt rates of the ice shelves derived from satellite remote sensing yield
a spatially heterogeneous as well as temporally variable distribution of basal melt (Sun et al., 2019). These ice
shelves are directly coupled to the regional ecosystem. In particular upwelling of sediment-laden plumes as part
of the overturning inside the ice shelf cavities may be a major supplier of nutrients, trace metals (iron and other
bio-essential elements), as well as inorganic and organic carbon. "Cold" ice shelf cavities are usually identified
by outflows of ice shelf water plumes that are colder than the freezing temperature at surface pressure, which
leads to platelet and potentially marine ice formation. Due to relatively fresh and hence buoyant continental shelf
water masses, formation of dense High Salinity Shelf Water is absent along the DML coast, which is the driver of
a vivid ice pump and marine ice formation in other regions of the Antarctic (Nicholls et al., 2009; Herraiz-
Borreguero et al., 2016). However, refreezing under the ice shelf (Hattermann et al., 2012), outflow of potentially
supercooled ice shelf water (ISW) (Nøst et al., 2011), as well as accretion of significant amounts of platelet ice
beneath coastal landfast ice (Arndt et al., 2020) have been observed, and these could play a major role in the
productivity of the whole Weddell Gyre (Kauko et al., 2021).
Over the recent decades, a slight increase in Antarctic sea-ice extent has been observed, with considerable spatial
and temporal variabilities (Parkinson, 2019), even though the summer sea-ice minimum has been below the long-
term trend in the past seven years, with a record low in February 2022 (Turner et al., 2022). However, the low
sea-ice extent period is not yet long enough to conclude a regime shift or a change in long-term trends. Overall,
the sea ice modulates surface momentum and buoyancy fluxes (Zhou et al., 2014) and affects the cycling of
nutrient and gas exchanges between ocean and atmosphere (Vancoppenolle et al., 2013).
The present-day atmospheric and oceanic $CO_2$ levels are projected to reach much higher values towards the end
of the century (Hoegh-Guldberg and Bruno, 2010; IPCC, 2022). As iron chemical speciation strongly depends on
$CO_2$ (Liu and Millero, 2002), iron seawater chemistry will be altered under high $CO_2$ concentrations (Ye et al.,
2020), with unknown effects for SO phytoplankton productivity (Pausch et al., 2022). Based on field observations,
the increase in atmospheric $CO_2$ has already turned the Weddell Sea from a $CO_2$ source into a $CO_2$ sink due to
elevated storage of $CO_2$ in the surface layer (Hoppema, 2004). However, the $CO_2$ exchange with the atmosphere
is not well quantified for the entire high-latitude SO due to the paucity of data, especially from the winter season
(Lenton et al., 2013). Warming induced strengthening of the subpolar westerlies (Thompson et al., 2011) has
caused stronger upwelling of carbon- and nutrient-rich deep water (Hoppema et al., 2015). It is presently unknown
if such vertical water transport will continue in the future. There is currently insufficient data for reliable
projections of the responses of Antarctic organisms to ocean acidification, together with changes in other
environmental factors, such as warming, light and nutrient availability (Seifert et al., 2020).
Primary production in the upper water column and the sea ice is regulated by an interplay of mostly climate-
sensitive environmental factors, including the seasonal sea-ice growth and melt, water-column stratification and
associated light regimes (Arrigo et al., 2008), as well as the availability of nutrients and trace elements, especially
iron (see e.g., McGillicuddy et al., 2015; Morley et al., 2020). In particular, trace metal data across the Weddell
Sea are still sparse, with evidence of low concentrations of both iron and manganese (Balaguer et al.; 2022).
Meteorological features (e.g., storms) can produce sudden and massive particle pulses that may cover hundreds
of square kilometres of the continental shelf with the sinking of tons of organic carbon in a few days (Isla et al.,
2009) and causing strong temporal variations in sub-ice shelf melting, thus increasing freshwater fluxes. Although
degradation processes in the upper water column are particularly intense in the Weddell Gyre (Usbeck et al.,
2002), the organic matter that reaches the seabed is sufficient to sustain diverse and abundant benthic
communities, which contribute substantially to the remineralization of organic matter (see also sections 3.2 and
3.3), especially in shelf regions (Brasier et al., 2021).
**3.1.2 Objectives**
To fill the current lack of understanding of observed and expected changes in the physical-chemical environment
and their impacts on biogeochemical fluxes in the marine ecosystem, we shall address the following objectives
(for closely related ecosystem services see section 3.3):

- Monitor the shelf-slope boundary current system and slope front structure to detect changes in the physical environment to (a) improve process understanding and develop links to ecosystem dynamics (b) assess the along-flow evolution and spatial connectivity of in- and outflow gateways of the eastern Weddell Sea;

- Understand basin-wide and climate-sensitive changes in ice-shelf/ocean interactions, such as spatio-temporal variability of basal melt rates underneath the Ekström Ice Shelf and production of platelet ice;

- Understand the atmosphere-sea ice-ice shelf variability and interaction, in particular drivers for pack ice and fast ice dynamics, seasonal surface evolution as an indicator of the under-ice light availability, and orographic changes of the cryosphere (e.g., ice shelf freeboard height), which can impact the marine ecosystems;

- Quantify key variables that structure the main ecosystem compartments including sea ice dynamics, water mass characteristics, as well as sea floor processes, to allow separating extrinsic (anthropogenic) from intrinsic (system-immanent) impact drivers;

- Assess carbon, nutrient, and trace-element cycling within and among these ecosystem compartments under current and future climatic conditions, to contribute to a better understanding of SO ecosystem functioning and their changes over time.

To integrate the above points into a holistic understanding of the physio-biogeochemical system, co-located and coordinated observations of multidisciplinary parameters are needed at a new set of distributed sites that are able to resolve the spatial connectivity in along-flow and across-gradient dimension. Also, existing long-term hydrographic, nutrients, iron, $CO_2$ and oxygen (and transient tracers) records must be continued (Fahrbach et al., 2011; van Heuven et al., 2011), while parameters and methods of measurement and analysis need to be reassessed and, where appropriate, be adapted to allow for the detection of drivers of change (e.g., the importance of buoyancy fluxes from air/sea-ice interactions and the role of ocean eddies for cross-shelf exchange). This approach will also enable the detection of abrupt and extreme events (e.g., storms) and periodic processes (e.g., tides; Isla et al., 2006) that may be essential for the overall energy and matter flow but are often overlooked.

The time-series measurements at the long-term observatory shall monitor inflow and outflow of the eastern WS boundary current to enable discrimination between meridional overturning, lateral advection processes, and water-mass transformations that connect local processes with the large-scale circulation. A quantification of the Antarctic Slope Undercurrent will help to determine its role in eastward transport of nutrients, trace elements, larvae, biotas, etc. beneath the westward-flowing Slope Current. On a basin-wide scale, these data shall complement the on-going ARGO float programme in the interior WS and serve as an upstream gauge for the recently established observatories at the southern WS continental slope/shelf, beneath the Filchner Ice Shelf, and at the Antarctic Peninsula. In particular, the Kapp Norvegia sub-area is a key location for observing the evolution downstream of the open ocean (Kauko et al., 2021), fast ice and under-ice shelf observatories at Fimbul Ice Shelf (Hattermann et al., 2012), and for monitoring changes that are expected to affect much of the southern WS. The IEAMaR stations in the coastal Ekström sub-area shall complement with on-going long-term observations of the fast ice-shelf ice-ocean interactions (including regular measurements of fast ice properties, the water column beneath and the basal/surface mass budget of the adjacent shelf ice) and PALAOA in Atka Bay and at Neumayer Station III, adding to the understanding of the impact of ice shelf-ocean interactions along the eastern coast of the

IEAMaR region. Moreover, Neumayer Station III shall provide an in-reach laboratory for process studies, in
particular when combined with long-term moorings beneath the ice shelf that are currently under development.
**3.1.3 Methods**
The major observing systems shall combine fixed moorings, drifting sea-ice observatories (e.g., Jackson et al.,
2013; Nicolaus et al., 2021), benthic observatories, autonomous underwater vehicles (including gliders), regular
ship-based observation work and satellite-based remote sensing. These will be complemented by atmospheric,
biological, cryospheric and oceanographic long-term observatory infrastructure at and in the vicinity of Neumayer
Station III. Long-term moorings shall be equipped with sediment traps, and sensors for monitoring transport,
bottom-water characteristics, WDW interface depth, and the upper ocean buoyancy budget, as well as
biogeochemical parameters, e.g. dissolved $O_2$ (e.g., Bittig et al., 2018), turbidity (e.g., Boss et al., 2015), $pCO_2$
(e.g., Lai et al., 2018), photosynthetically active radiation, and pH (e.g., Okazaki et al., 2017), fluorescence (as a
proxy for chlorophyll and phytoplankton abundance) and other in-situ tracers. Optical sensors will be used to
measure nitrate (e.g., Sakamoto et al., 2017) and Colored Dissolved Organic Matter. It is also envisaged to make
use of the promising recent development of Lab-on-Chip sensors for assessing Dissolved Inorganic Carbon, pH,
nitrate, phosphate and iron (Nightingale et al., 2015). Active acoustic techniques shall be used to determine depth
profiles of currents, zooplankton and fish abundance. In addition, bio-optical platforms equipped with particle
cameras and gel traps shall collect sinking particles to assess the flux of organic matter from the mixed layer to
the seafloor.
For sea-ice monitoring, upward looking sonar and acoustic Doppler current profilers (thickness and ice drift
velocity) at the backbone moorings shall be combined with fixed electromagnetic induction (EM) stations (Brett
et al., 2020) and monitoring of optical properties and relevant biogeochemical properties through fast ice. Drifting
ice-tethered autonomous observatories will provide data on biogeochemical properties of sea-ice and the water
column, zooplankton and fish distribution (see also section 3.2.3) during their drift through the Weddell Gyre.
Repeated airborne EM and broadband radar grids (Haas et al., 2021; Jutila et al., 2022) shall provide coincident
snow and ice thickness and roughness information to characterize ice regimes of first- and second year sea ice of
different origin. Ice/ocean buoys shall provide year-round time series of meteorological parameters and air/sea-
ice/ocean interactions in a larger geographical context, covering an extended set of ECVs (Lavergne et al., 2022),
as well as ocean-surface stress and ocean-surface heat flux to support the cross-disciplinary concept of the
IEAMaR long-term observatory.
Open-ocean observations of cryospheric components will be complemented by sea-ice coring for direct
physical, biological, and chemical material collection and direct data measurements should be conducted in all
three sub-areas, primarily in Atka Bay, and preferably at the same or similar locations and at the same time of
year on a regular basis from the ship, with helicopter support if necessary. Sampling shall be done on different
ice types, including fast ice, seasonal ice, snow cover and platelet ice, where these exist, together with sub-sea
ice ocean properties from manual CTD casts (Arndt et al., 2020), The collection of oceanographic data by a
mooring below the Ekström Ice Shelf since 2005 should be continued and extended. It is well protected from
the regular ice-berg traffic in front of the shelf (Oetting et al., 2022) and safer than those hung from the ice-
shelf edge, which is subject to regular calving events. The mooring holds a passive acoustic recorder and a
CTD and has been operational until February 2022 when the iceshelf broke off and the cables were torn. This

long-term data series is extremely valuable for understanding physical processes at the sub-surface of the ice shelf, as well as coastal oceanographic processes, including the dynamics of pelagic species composition on a year-round basis and in relation to the environment. Due to the service work carried out by the overwinterers of the Neumayer station III it proves the potential longevity of such set ups - compensating for the cost of installation - which could never have been achieved by moorings deployed in front of the ice shelf where there is heavy iceberg traffic.

Satellite-based remote sensing shall be used to determine sea-ice variables (extent, concentration, thickness, snow cover, drift, age, surface temperature, surface albedo), sea-surface temperatures, glaciological variables (surface elevation, ice-flow velocity, basal melting, see e.g. Berger et al., 2017; Eisen et al., 2020) surface and total mass balance (e.g. Eisen et al., 2019), calving, the spatio-temporal distribution of the biomass of pelagic primary producers (chlorophyll), and to systematic monitoring of seal and penguin abundances and distributions.

Shipborne air-chemistry and autonomous remote particle concentration measurements are also desirable. They can be tied to the data from the Neumayer air-chemistry observatory (Weller et al., 2006). Ship-board measurements will also allow the ground-truthing of the automated systems, ad-hoc experiments and field studies. The latter encompasses CTD transects and concurrent sampling for radiotracers, trace metals and nutrients.

At the seabed, observations and samplings shall be conducted with grabs and corers to assess benthic-pelagic coupling processes (e.g., seasonal deposition and degradation of labile organic matter), sediment redox cycling, and early diagenetic processes, using rate measurements, as well as biomarker and stable isotope analysis. Sediment traps shall enable the determination of the dynamics of sinking rates and the relative importance of different types of particles at various temporal scales. Benthic geochemical observatories shall complement the oceanographic moorings, equipped with a similar suite of sensors to monitor conservative and reactive compounds in the nepheloid layer and currents. Benthic oxygen fluxes shall be determined using eddy-covariance technology and repeated sediment $O_2$ profiling. For additional methods to acquire biological data in the context of the flux of energy and biomass see Section 3.2.3.

## 3.2 Organisms and ecosystems – adaptations, biodiversity and ecosystem functioning

### 3.2.1 Background

The projected sea-ice decline and water-column changes (Moline et al., 2004; Trimborn et al., 2017; Eayrs et al., 2021) will profoundly affect primary production and zooplankton composition, with cascading but so far largely unknown implications for the entire SO food web, including top predators and the benthic system (Atkinson et al., 2019; Hill et al., 2019; Steiner et al., 2021). In general, all organisms from different trophic guilds respond to environmental changes by migration or extinction unless they can acclimatise because of phenotypic plasticity and genotypic adaptation through natural selection (Somero, 2012). The underlying genetic architecture of organismic adaptation is responsible for shifts in the ecological niche width of a species under changed conditions. Comprehensive process studies, which relate transcriptomic/proteomic responses and threshold temperatures to long-term ecophysiological parameters including growth performance (Windisch et al., 2014), are lacking so far for key species in the high-latitude SO. Such studies would contribute to addressing the general long-term objective of establishing the missing link between genotype and phenotype (Oellermann et al., 2015) and of

understanding the role of population structure and temporal variation for the adaptability to environmental change
(Lancaster et al., 2016).
The adaptation of single species to environmental change, and, therefore, shifts in their physiological and
behavioural performance, has consequences for species interactions, biodiversity and functioning of communities
(Gutt et al., 2018). This includes competition, predator-prey relationships and responses to disturbances including
extreme events in sea-ice dynamics, iceberg calving and scouring, spatio-temporal shifts in water masses or
weather-driven mass occurrence of phytodetritus at the sediment surface (Sañé et al., 2012).
It is generally known that biodiversity drives ecosystem functioning, such as productivity, energy transfer and
remineralization (Naem et al., 2012), as well as ecosystem stability. So far, the biodiversity-ecosystem functioning
(BEF) relationship and its climate-sensitivity are virtually unknown for high-latitude SO pelagic and benthic
systems, although such knowledge is essential to understand and predict developments in ecosystem structure and
function in response to climate and other environmental change. Also, our knowledge of whole-community
vulnerability or robustness is still very poor, especially for the slow-growing and immobile epibenthos (Gutt et
al., 2018), which cannot respond to rapid environmental changes with immediate shifts in spatial distribution (Isla
and Gerdes, 2019) but provides with its three-dimensional architecture specific micro-habitats for a rich associated
fauna.
The Antarctic sea-ice itself provides the habitat for a variety of unique taxa that contribute significantly to carbon
flux and nutrient cycling in the SO (Monti-Birkenmeier et al., 2017; Steiner et al., 2021). Ice algae are an important
source of carbon for pelagic and benthic communities (Meiners et al., 2018). Thus, the sea-ice cover critically
controls ecosystem functions and services in the Weddell Sea. In addition, ice-associated biotas play an important
role for the winter survival of various zooplankton taxa (Schaafsma et al., 2017; Kohlbach et al., 2018), and the
sea-ice habitat constitutes an important shelter and nursery ground for Antarctic krill (Meyer et al., 2017; David
et al., 2021). At the same time, sea-ice, its associated communities, related biogeochemical processes and trophic
interactions are highly sensitive to climate-induced changes in structure, temporal dynamics and spatial extent.
Population sizes have been estimated for emperor penguins (*Aptenodytes forsteri*) (Fretwell et al., 2012) and
Weddell seals (*Leptonychotes weddellii*) (LaRue et al., 2021), based on the analysis of satellite images. However,
for most Weddell Sea meso- and top-predators, population sizes are still unknown (Gurarie et al. 2017; Richter et
al. 2018), which limits our ability to assess population health and trends, as well as predator responses to climate
change. Filling this knowledge gap is especially important because the wider IEAMaR area is generally known as
a likely important foraging ground for several Antarctic seal and penguin species (McIntyre et al., 2012; Bester
et al., 2020; Wege et al., 2021a).

### 3.2.2 Objectives

The time-series observations of biodiversity and ecosystem variables, to be conducted in parallel with physical-
chemical parameters (see Section 3.1), will address the following objectives:
• Identify key species, assemblages and functional groups (covering a range of ecologically important, trophic
levels, habitats, as well as traits, such as population size, reproduction, mortality, growth rates, and
competition) for monitoring and preparation for targeted LTER work;

- Assess the adaptive and acclimatory scope of ice-associated, pelagic and benthic key species: genetic diversity, ecophysiological plasticity, adaptive strategies and capacities, spanning the whole life-cycle over several generations;
- Determine changes in spatial distribution of selectively neutral and adaptive alleles (e.g., stress response) in populations of vulnerable species (genomics, transcriptomics);
- Determine changes in species spatial distribution and foraging habitats compared to known distributions and calculated Areas of Ecological Significance (Hindell et al., 2020);
- Evaluate taxonomic and functional biodiversity of ice-associated, pelagic and benthic biotas, encompassing a wide range of organisms from microbes to top predators, and identify Areas of Ecological Significance based on species others than top predators;
- Identify and understand relationships between biodiversity and ecosystem functioning, including climate feedbacks, such as energy flow, production, remineralization and species interactions;
- Assess robustness or vulnerability of ecosystems in the IEAMaR region and the impact of multiple drivers with respect to anthropogenic changes in biodiversity (including the establishment and spread of non-indigenous species).

Comprehensive knowledge on the structure and functioning of genes is the basis to assess the role of individuals in their population, community or ecosystem. In cases where the entire species-specific ecophysiology cannot be studied (yet) by biomolecular "-omics" studies, whole-organism in-situ or in-vitro experiments can provide valuable insights (Strobel et al., 2012). Therefore, the long-term IEAMaR work will survey ecophysiological parameters, gene expression and life cycles of ecological key taxa, such as phytoplankton, crustaceans (e.g., copepods, amphipods, isopods, euphausiids), fishes (mostly notothenioids), echinoderms, molluscs and selected sessile suspension feeders (e.g., sponges, ascidians, cnidarians, and bryozoans). All such studies aim to determine the environmental plasticity of single organisms and species with their intra- and inter-specific variability and validate the results from experiments or single-species models (Somero, 2012). The results will allow for understanding the complex environmental conditions under which organisms can persist or become locally extinct.

A reliable ice cover is crucial for the reproduction of Antarctic krill, ice-breeding pinnipeds and emperor penguins, and as a potential winter retreat for some species, e.g., Antarctic minke whales (Meyer et al., 2017; Filun et al., 2020). Polynyas also play a major role as foraging areas (e.g., Malpress et al., 2017; Labrousse et al., 2019). For other meso- and top-predator species, availability of ice-free surface for breeding and access to productive foraging grounds are key long-term population drivers (Younger et al., 2016). The logistical challenges of systematic long-term in-situ data collection limit our understanding of habitat use by top predators and their prey for many parts of the SO, including the continental slope areas that are home to adult Antarctic toothfish. The cross-disciplinary character of the IEAMaR observatory allows the combination of remote-sensed population assessments and continued studies of distribution, foraging ranges and behaviour as well as passive acoustic monitoring studies of SO top predator species related to key environmental features (e.g., Van Opzeeland et al., 2010; Thomisch et al., 2016; Hindell et al., 2020; Houstin et al., 2021; Oosthuizen et al., 2021; Schall et al., 2021; Wege et al., 2020, 2021a and b).

The BEF relationship shall be investigated in detail to assess ecosystem stability versus vulnerability for the given
biodiversity. For instance, analyses shall be carried out on whether a possible decline in species richness affects
primary production, energy transfer and nutrient recycling through changes in functional redundancy at a
community level. An increase in ecosystem functions (e.g., primary and secondary production) with decreasing
biodiversity are expected if fast-growing species (e.g., "pioneers") become dominant. Regional biodiversity may
also increase with a shift towards less polar conditions, if sub-Antarctic species (e.g., Patagonian toothfish)
immigrate into the WS displacing native high-Antarctic benthic species (e.g., Antarctic toothfish or *Trematomus*
spp.) (Griffiths et al., 2017). However, high-Antarctic species are often eurybathic in their depth distribution and
thus may preserve their climate envelopes by migrating to deeper waters (see Barnes and Kuklinski, 2010), a
process that may be facilitated by boosted pelatic primary production due to sea-ice losses (Arrigo et al., 2008).
Mobile pelagic species are foraging and travelling further southward from sub-Antarctic island colonies to forage
at the ice edge of Southern Ocean waters, increasing competition potential with Antarctic species (e.g., Cristofari
et al., 2018; Krüger et al., 2018; Reisinger et al., 2022a). Addressing these objectives demands investigations of
patterns and processes of biodiversity in all their facets, such as species richness, evenness, functional diversity,
dispersal, reproduction (including brood care of icefish; Purser et al., 2022), recruitment, growth and mortality, as
well as abundance and biomass. Analyses of such species-specific key traits shall be linked to "first-principle"
process studies to understand the relationship of the sympagic, pelagic and benthic communities, including apex
predators, with ecosystem functions and services (see also Section 3.3). Moreover, the effect of different spatial
scales shall be taken into account, since reduced local biodiversity can lead to a higher spatial species patchiness
and higher temporal species turnover with yet unknown consequences for ecosystem stability, resilience and
function.

### 610 3.2.3 Methods

Information on key taxa shall be collected across various spatial scales at the IEAMaR observatory and adjacent
pelagic transects (Fig. 1) by means of various methodological approaches and in parallel to the flux studies (see
Section 3.1). Phytoplankton and pelagic primary consumers, such as krill, copepods and young fish larvae, as well
as secondary consumers, such as Antarctic silverfish *Pleuragramma antarctica*, shall be studied by CTD and
rosette casts and pelagic net catches, to assess species composition, abundance, population parameters and feeding
condition, and compared with data provided by acoustic systems. Benthic surveys shall primarily be conducted
by means of minimally invasive methods (e.g., traps, corers, autonomous seabed and under-shelf ice sampling
and acoustic as well as optical imaging, and scientific long-line fishery), to minimize the anthropogenic impact of
invasive sampling methods (e.g., bottom trawls).
Higher-order predator studies shall be continued by the instrumentation of animals with CTD-satellite trackers
and biologging devices (e.g., Nachtsheim et al., 2019; Houstin et al., 2022), as well as physiological and nutritional
studies. Population estimates and habitat distribution (e.g., LaRue et al., 2021; Wege et al., 2021b) of seal and
penguin populations shall be monitored using a combination of airborne and Very High Resolution (VHR) satellite
imaging, including autonomous year-round observations (Richter et al., 2018; Fretwell and Trathan, 2020).
Images will focus on locations representative for the core stations established at regular intervals. Both biologging
data and VHR imagery data shall be used to determine critical habitats. As more data become available, we can
project distribution changes of these core habitats into the future using climate modelling (e.g., Reisinger et al.,

2022a). Oceanographic conditions at emperor penguin foraging hotspots shall be studied using autonomous underwater vehicles that can be deployed from Atka Bay by actively following satellite tagged specimens. Passive acoustic monitoring data shall be used for larger spatio-temporal scale soundscape studies (Menze et al., 2017) and to investigate how marine mammal occurrence relates to fluctuations in their ice-dominated habitats.

However, recurring sampling and archiving of organisms suitable for molecular analysis (e.g., every five years) shall also take place to ensure ground-truthing of non-invasive methods. A concept shall be developed, which allows a sound identification of ecological key species or functional groups and addresses the limitations of resources (time, taxonomic expertise, sorting effort). Using molecular (meta) barcoding, cryptic species shall be identified. Imaging surveys shall allow for detecting shifts in benthic community composition, species traits, interactions, diversity, biomass and size structure of populations. Existing semi-automated analyses of such images by deep-learning networks (e.g., Schöning et al., 2012) shall be adapted and improved for analysis of SO benthos. The analysis of eDNA shall allow for detecting whole-community changes in biodiversity. Quantitative information on all benthos fractions is important to separate short-term remineralization from long-term burial of carbon and other nutrients.

Once observational and analytical baselines have been established, advanced experimental field studies (e.g., in situ respirometry, http://www.mbari.org/emerging-science-of-a-high-co2low-ph-ocean-deep-water-foce/, last access: 23 August 2022) and long-term video observations of local key species and assemblages within their natural habitats and a focus on interactions shall follow. In combination with on-site laboratory experiments (e.g., at the Neumayer Station III), they will help to unravel life-history strategies, life stage-specific spatial and temporal distributions and their adaptive scope over generations. Furthermore, internal and external data loggers suitable for smaller marine organisms, such as fish and invertebrates, audio-visual loggers to study foraging behaviour of predators, long-term tracking of their preferred water temperature and depth, but also of physiological parameters, such as heart rate, blood flow and tissue oxygenation, have come into reach. Semi-permanent moorings and lander systems (see Section 3.1) shall serve as a (power) base for those applications and allow for controlled deployment of traps and other gear. Otoliths of fishes shall be used as an archive of temperature preference and utilized resources in fishes.

Autonomous bio-environmental observatories shall constitute an important pillar of the long-term IEAMaR observatory. By combining multiple sensors on bottom-moored, sea-ice-moored and free-drifting platforms, the spatio-temporal gaps between field campaigns will be closed with high-resolution data. These systems should merge existing state-of-the-art environmental sensors, such as CTDs, nutrient sensors, fluorometers, spectral radiometers and optical sediment traps, with the newest technology to monitor organisms beyond microbes. Examples for such sensors are camera systems, autonomous multi-frequency echosounders (e.g. Acoustic Zooplankton and Fish Profilers, Wideband Autonomous Transceiver), which are able to record and transmit data and to receive real-time manipulation of the sampling programme, as well as automatic eDNA samplers and imaging profilers (e.g. Underwater Vision Profiler).

The pelagic community will be sampled with an ultra-clean CTD and under-way filtration systems for phytoplankton, ship-mounted echosounders, Rectangular Midwater Trawls for plankton and nekton and multinets and imaging zooplankton profilers (e.g., Lightframe On-sight Key Species Investigations, LOKI) for the mesofauna (e.g., Schnack-Schiel et al., 2008; Flores et al., 2014). Net catches will be used to ground-truth biomass

and community data derived from continuous sampling with multi-frequency broadband echosounders as well as
from autonomous observatories equipped with echosounders. The community under the sea ice will be sampled
using under-ice trawls, alongside with physical parameters of the sea-ice and underlying water (Castellani et al.,
2022), and the sympagic in-ice community composition will be investigated on ice stations by ice core sampling
(for sampling strategy see section 3.1.3) using both molecular and morphological techniques (Miller et al., 2015,
Monti-Birkenmeier, 2017). Trophic relationships, including match-mismatch phenomena, are to be studied by a
variety of methods, such as, analyses of gut content and of tissues for lipid composition and isotope ratios, of
gonads for maturity, and entire specimens for body conditions.
**3.3 Ecosystem services and human impacts**
### 3.3.1 Background
Ecosystem services (ES), i.e., the benefits that people obtain from ecosystem functions, also called *Nature's*
*Contributions to People* to compensate for negative effects (Díaz et al., 2018), have received increased attention
from stakeholders during recent years. The ES framework aligns economic considerations with nature
conservation and thereby addresses diverse and powerful questions (Simpson, 2011). However, the quantification
of ES is one of the greatest challenges of current ecosystem science (Burkhard et al., 2012), especially due to the
spatial and temporal variability of ecosystems, particularly in the marine domain (Barbier, 2007). This challenge
is aggravated by the fact that many seascapes are under-represented in global assessments (e.g., TEEB, 2012),
including the SO that has not yet been subject of any detailed regional ES assessment (Grant et al., 2013). The
fact that most ES provided by the oceans, particularly remote marine areas such as the SO, seldom have on-site
beneficiaries (for instance, markets for Antarctic fisheries products, such as toothfish, are mainly in Japan and
North America (Catarci, 2004)) adds to the complexity of the topic. Moreover, the introduction of a payment for
ecosystem services (PES) has been discussed for Antarctic tourism (Verbitsky, 2018).
Regulating ES provided by the SO including the WS are also beneficial to human populations on a global scale,
e.g., regarding climate regulation, sea-level rise, carbon sequestration, oxygen production, remineralization of
organic matter, and natural genetic heritage and biodiversity (Deininger et al., 2016; Pertierra et al., 2021; Steiner
et al., 2021). The long-term IEAMaR concept shall primarily contribute to a better understanding of core
ecosystem functions and services regarding two aspects: 1) improving carbon sequestration budgets and 2)
contrasting direct human impacts (fishing) and nature conservation efforts.
A meta-analysis revealed that ocean acidification could negatively affect autotrophic organisms, mainly
phytoplankton, at $CO_2$ levels above 1,000 µatm and invertebrates above 1,500 µatm (Hancock et al., 2020). Hence,
Antarctic organisms are likely to be susceptible to ocean acidification and thereby likely to change their
contribution to ecosystem services in the future. The SO, especially the coastal parts, is potentially a strong sink
for anthropogenic carbon (Arrigo et al., 2008). However, it is also a highly dynamic and heterogeneous region
(Gutt et al., 2013a; Tagliabue and Arrigo 2016; Jones et al. 2017) that is poorly sampled in large areas (Arrigo et
al., 2015). The supply of iron is considered to control how much $CO_2$ is biologically fixed by phytoplankton
photosynthesis. There is, however, a lack in knowledge on the magnitude and the importance of different iron
sources on phytoplankton productivity, including melting of sea ice and icebergs, dust deposition and iron
recycling by different grazers, with changes to be expected in the future (Trimborn et al., 2017; Böckmann et al.,

2021). Furthermore, models of the export production and $CO_2$ uptake disagree on the processes that lead to the export of organic carbon today, let alone in the future (Laufkötter et al., 2016). These deficiencies in our understanding of biological processes induce large uncertainties in the projections of future primary production (Frölicher et al., 2016) and the oceanic carbon sink, thus hindering the quantification of an important ES from the SO. Research on carbon sequestration is closely linked to research on ecosystem functioning; as a result, this research theme overlaps partially with Section *3.1*.

The WS is the last SO area where no or only limited fishing has taken place to date. Commercial krill fisheries are concentrated around the Antarctic Peninsula and the Scotia Sea, where krill abundances are much higher than in the southern and eastern WS (Atkinson et al., 2019). Over almost the last 20 years, longline exploratory fishing for Antarctic toothfish has been carried out on the continental slope in the CCAMLR Statistical Area 48.6, i.e., off the ice shelf where Neumayer Station III is located and further eastwards (Teschke et al., 2016). Adult Antarctic toothfish are demersal top predators, which can grow to over 2 metres in length and reach over 50 years in age. The climate-sensitivity of these fish populations (Cheung et al., 2008) and of the marine ecosystems and food webs they are part of is a main reason for the plan to establish a Marine Protected Area (MPA) in this region (Fig. 5, for the scientific justification of the eastern WSMPA phase 2 area see Lowther et al., 2022). Regarding conservation of biodiversity, special attention has to be paid to rare habitats, since they are especially vulnerable. Example are poorly researched polar marine habitats, such as the underside of ice shelves and floating glacier tongues or the unusually shallow and especially diverse Norsel Bank in the Kapp Norvegia sub-area. A hierarchical classification of benthic biodiversity has been carried out in the context of assessing protected areas in the Southern Ocean (Douglass et al., 2014), but in general, an understanding of the mechanisms driving observed or projected changes remains largely unknown, including the role of the relatively high benthic biodiversity for the stability of the entire system.

### 3.3.2 Objectives

The objectives to quantify carbon sequestration as a major ecosystem service and facilitate the conservation of ecosystem functioning and services of the WS and DML coast are as follows:

- Quantify the carbon sink, its change, and drivers and temporal change in the IEAMaR region by analysing oxygen production through primary production and the biologically- and physically-mediated transport of carbon from the ocean surface to seabed sediments;
- Develop a robust understanding of biogeochemical processes from multidisciplinary high-resolution time-series data, which may also be used for model evaluation and development;
- Identify key taxa of carbon and nutrient transfer, especially for carbon export, storage and remineralization, to improve future climate scenario projections;
- Develop strategies to protect species assemblages based on the knowledge of key species and rare, unique, highly diverse or endemic habitats  (including essential habitats for top predators);
- Provide the scientific basis to protect environmental features and species (including their populations and life stages) on various geographical scales, which are key to the functional integrity and viability of regional ecosystems processes;
- Establish scientific reference areas to monitor the effects of climate change, fishing and other human activities;

•  Protect potential refugia for, inter alia, top predators, fish, other ice-dependent and highly cold-adapted and

744        sympagic species, to support their resilience and ability to adapt to the effects of climate change.

The long-term IEAMaR efforts shall provide the opportunity to study the year-round carbon flux into and out of
the mixed layer in relation to meteorological (e.g., wind) and biological drivers (e.g., primary production and
composition, abundance, growth, metabolism, as well as mortality of key grazers). They shall thus contribute to
providing a circumpolar assessment of the biological carbon sink structure and the sequestration of $CO_2$ from
the atmosphere for hundreds to thousands of years, including baseline and variability in carbon capture, storage
and sequestration components by the sinking of faeces and phytodetritus, as well as potential changes in the
eastern WS region (such as climate change feedback strength). Moreover, they shall allow for separating the
physical from biological processes that lead to a transfer of carbon to the ocean interior and seabed. The carbon
flux to the sea floor further determines the redox state of sediments and is positively correlated with the efflux of
nutrients and iron (Graham et al., 2015). This positive feedback is pronounced in shallow shelf seas where
vertical pathways are short and pelagic-benthic coupling triggered by sedimentation events is enhanced. At least
some benthic suspension feeders have significantly increased benthic carbon and silicate storage on the seafloor
of the wider IEAMaR region over the last two decades (Gutt et al., 2013b; Barnes, 2015) and contribute to the
remineralization of organic matter to be further quantified. IEAMaR data shall provide a sound basis for the
development and validation of regional biogeochemical carbon budgets and models. They will also allow for the
assessment of changes in the efficiency of carbon export to deeper waters and hence benthic carbon supply, as
well as of the fate of other nutrients and the potential carbon sink role of the long-term IEAMaR sites that are
representative for the mostly ice-covered high latitude SO (see also Section 3.1).



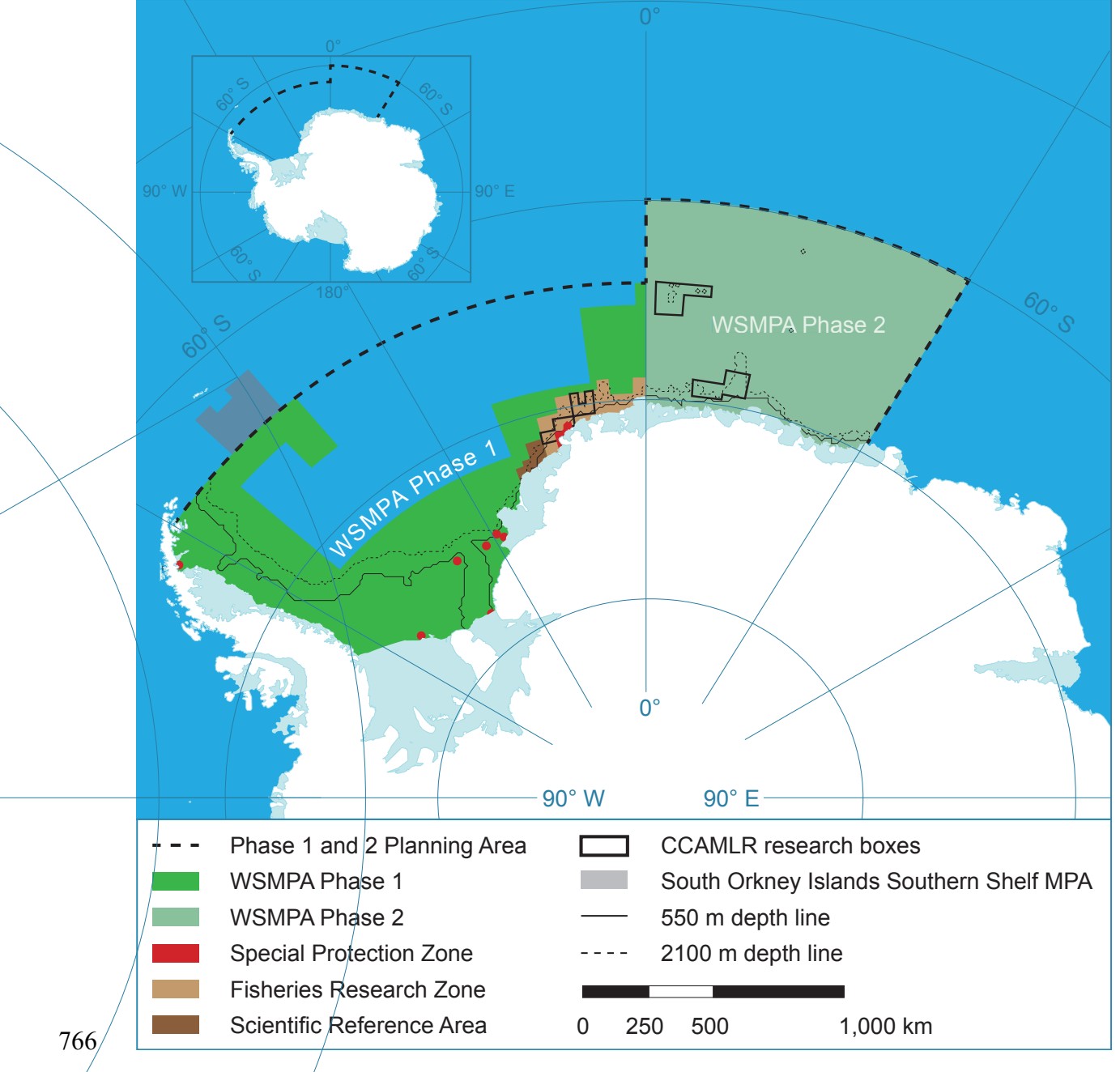

**Figure 5:** Proposed regions for a Weddell Sea Marine Protected Area (WSMPA) Phase 1 and Phase 2. Changed after: AWI Factsheet Weddell Sea: Eight Reasons for a Marine Protected Area (https://www.awi.de/fileadmin/user_upload/AWI/Ueber_uns/Service/Presse/2016/4_Quartal/KM_Weddellmeer_MPA/WEB_DE_Factsheet_Weddellmeer.pdf, last access 23 August 2022). Map design by Yves Nowak, AWI.

With reference to the proposed WSMPA, the IEAMaR observatory shall provide the focal point for the research and monitoring activities required in a WSMPA management plan and is important for the regular review of the effectiveness of the WSMPA. To ensure synergistic effects for science and marine conservation policy, the location of the observatory is partly congruent with the proposed WSMPA (Fig. 5). After the adoption of the WSMPA by CCAMLR, the long-term IEAMaR work shall provide research-based long-term data on the natural

development of the protected environments and biotas. The sympagic, pelagic and benthic habitats to be
monitored because they are at least partly representative for the East Antarctic SO rare and especially vulnerable
habitats include the shelf-ice associated cryo-benthos (Watanabe et al., 2006; Gutt and Dieckmann, 2021) and
benthic communities at unusually shallow sites that are thus exposed to unusual environmental conditions and
disturbances (Raguá-Gil et al., 2004). LTER work at the IEAMaR observatory shall also provide insight into the
biology, life cycle and trophic role of Antarctic toothfish. Since 2014, every year around 200 tonnes of Antarctic
toothfish are caught by fishing vessels operating in the CCAMLR research block 48.6_5 in front of the DML
coast (Fig. 5). Combining the information obtained by these longline operations with data obtained by the
IEAMaR will allow comparisons of the benthic habitats in this research block with similar habitats in unfished
areas to study the physical disturbance and effects of longline fishing. The close proximity and partial overlap of
the IEAMaR region with the CCAMLR research block provides also the opportunity to carry out further studies
of local and regional food web and ecosystem effects caused by the annual removal of large quantities of *D.*
*mawsoni* as a top demersal predator. Less consumption will certainly have impacts on its prey species and the
entire seabed community. In addition to this predation release effect, changes to toothfish abundances caused by
fisheries could also impact toothfish predators like Weddell seals and whales (sperm, killer and Arnoux beaked
whale).
**3.3.3 Methods**
During the long-term observations and further complementary scientific work at the IEAMaR observatory, sea-
surface $pCO_2$ in a coastal Antarctic region shall be assessed with high resolution. A $pCO_2$ sensor (Lai et al., 2018)
will be part of a novel mooring design, which is protected against iceberg scouring and shall be deployed in a
synoptic approach in combination with other instruments (see Sections 3.1.3 and 3.2.3 above). Based on these
combined data, the physical and biogeochemical carbon transport shall be based on the crossdisciplinary approach
described in the sections above. The continuous time series at high-temporal resolution (hourly for sensors,
biweekly for sediment traps) shall be used for the evaluation of global and regional biogeochemical models. The
gained process understanding shall be used to improve parameterizations of biogeochemical processes in models,
e.g., the mechanisms that lead to the formation and disaggregation of sinking particles. Sediment-core studies
shall allow monitoring benthic remineralization rates and nutrient efflux in relation with benthic fauna
composition, enabling the estimation of changes in the upward mixing of essential nutrients.
Once the Weddell Sea MPA has been approved, the development and implementation of a detailed research and
monitoring plan is a task for the CCAMLR members. The then required WSMPA research and monitoring
activities would be carried out in the IEAMaR area, making use of or being supported by the observatory
infrastructure (see Sections 3.1 and 3.2), e.g., by monitoring the temporal variability of benthic fauna with time-
lapse cameras. In addition, scientific sampling of the benthic and pelagic fish fauna shall be carried out in areas
designated for this purpose. These studies will be designed to complement on the one hand the results of historical
fish research carried out in the 1980s and 1990s, e.g., by the Alfred Wegener Institute, and on the other hand the
data on toothfish, toothfish prey and by-catch species obtained in the commercial long-line operations in the
CCAMLR fisheries research block 48.6_5 (Fig. 5). This will contribute to both the development of a stock
hypothesis of Antarctic toothfish in the larger Weddell Sea area and the research and monitoring required by the
Weddell Sea MPA proposal. Advanced spatially explicit and dynamic ecological modelling that includes biotic
and abiotic interactions will allow for an assessment of the effects of disturbances and environmental changes on
ecosystem functions and services. A major challenge in such ecological modelling is that a spatial resolution must
first be found that takes into account on the one hand the limitations of physical projections in downscaling
approaches and on the other the small-scale nature of biological patterns.

## 4 Added value

### 4.1 Lessons learned from previous long-term studies in the Southern Ocean

Repeated sampling over decades off the West Antarctic Peninsula and in the Atlantic sector of the SO showed
that Antarctic krill stocks (*Euphausia superba*) experienced climate-induced reductions; they partly shifted
southward and have partly been replaced by salps (mainly *Salpa thompsoni*; Atkinson et al., 2019; Hill et al.,
2019). However, the pelagic ecosystem off the western Antarctic Peninsula is different in that the shelf extends
much further than in the East Antarctic SO investigation area. Therefore, it offers more iron sources and leads to
an overall more productive ecosystem. In addition, the coastline is much more irregular and provides a special
habitat heterogeneity. The Palmer LTER was established in this area in 1990 (Smith et al., 2013), at a time when
climate was already changing rapidly in this Antarctic region. Surveys identified changes in pelagic food webs
west of the Antarctic Peninsula (Ducklow et al., 2006), where benthic inshore biodiversity has partly increased as
a result of long-term glacier retreat at King George Island (Sahade et al., 2015; Zwerschke et al., 2022). However,
the trends detected in primary production and for higher trophic levels are inconsistent, largely due to the
heterogeneity in sea-ice dynamics that in turn depend on variable meteorological conditions (e.g., Montes-Hugo
et al., 2009; Lin et al., 2021), with consequences for oceanic $CO_2$ uptake (Brown et al., 2019). The early
establishment of the Palmer LTER and additional studies covering the area from the northern tip to the southern
based of the Antarctic Peninsula allowed researchers to detect the impacts of climate change on various marine
ecosystems, highlighting the importance of establishing the IEAMaR observatory as early as possible, before the
onset of profound climate-change effects in the East Antarctic SO.
In the high-latitude East Antarctic SO, some extremely rare repeated surveys after several years provided insights
into an unexpectedly rapid recruitment and growth but also mass mortality of sponges and/or ascidians, e.g., in
McMurdo Sound (Ross Sea) (Dayton et al., 2013; Kim et al., 2019) and in the western Weddell Sea off the Larsen
ice shelves (Gutt et al., 2011). These findings were related to changing phytoplankton bloom dynamics triggered
by ice-shelf disintegration and calving icebergs in combination with altered sea-ice dynamics and iceberg scouring
impacts (Gutt and Piepenburg, 2003; Cape et al., 2014; Dayton et al., 2019). New blooms and benthic growth
spurred by regional ice shelf losses can create new carbon sinks, with corresponding feedback ramifications for
the climate (Peck et al., 2010; Barnes et al., 2018).
Off Dronning Maud Land, along the Prime Meridian, and in the Weddell Sea within the wider IEAMaR region,
but also elsewhere within the East Antarctic SO, long-term observations have documented a warming trend in
deep water properties (Smedsrud, 2005; Strass et al., 2020). The causes remain unclear (Fahrbach et al., 2006).
More recently, studies of the vertical (Cisewski and Strass, 2016) and horizontal ecosystem structure (Kauko et
al., 2021), together with the installation of a multidisciplinary moored ocean observatory along a shelf-slope
transect at 6° E (de Steur et al., 2019), revealed large interannual variability of phytoplankton blooms in the region.

Marine soundscapes of biological and physical origin have been monitored continuously since 2008 by the HAFOS network and the marine soundscape monitoring throughout the Weddell Sea area near the Neumayer Station III by the "Perennial Acoustic Observatory in the Antarctic Ocean" (PALAOA). These data revealed rich marine mammal communities that fluctuate in composition throughout the year and are sensitive to environmental anomalies that may increase in frequency under future climate conditions (e.g. Schall et al. 2021; Roca et al., in press). In addition, the long-term observations of the emperor penguin colony at Atka Bay provide continuous ground-truth calibration data for satellite remote sensing-based pan-Antarctic emperor penguin census studies (Richter et al., 2018). They also shed light on emperor penguin behaviour at sea, and showed that juvenile emperor penguins spend the majority of their time outside of proposed and existing Marine Protected Areas and venture as far north as the Antarctic Circumpolar Current (ACC), about 2000 km away from their breeding colony. These findings demonstrate that conservation efforts in confined to the SO proper are insufficient to protect emperor penguins. Because of the low fecundity of emperor penguins, the successful recruitment of juvenile cohorts is critical for emperor penguin population dynamics (Houstin et al., 2021). Off Coats Land and off western DML southwest of Atka Bay, investigations on the benthos carried out at irregular intervals over three decades indicate trends in taxonomic composition and traits (Pineda-Metz et al., 2020), which are superimposed by variations in sampling approaches and a pronounced small-scale heterogeneity (Gutt et al., 2013a).

The circumpolar "Retrospective Analysis of Antarctic Tracking Data" of SCAR highlighted Areas of Ecological Significance based on tracking data from 17 SO bird and mammal species over the past 30 years (Hindell et al., 2020; Ropert-Coudert et al., 2020). The study also predicted a long-term net loss of about a tenth of the Areas of Ecological Significance by 2100. The habitat-use of these predators indicates biodiversity patterns that require adequate representation in SO conservation and management planning (Reisinger et al., 2022b).

## 4.2 International integration

The ecological complexity to be tackled by the IEAMaR concept both demands and provides the potential for extensive international collaboration. The planned IEAMaR long-term observatory will also complement the work of similar observatories in the maritime Antarctic (e.g., Palmer LTER off the western Antarctic Peninsula) and in other regions in the high-latitude SO. Inter-comparability of methods and data among various LTER efforts shall be a priority in the implementation of the planned observation and activities in the IEAMaR region. In addition, this observatory can serve as a showcase project within the Southern Ocean Observing System (SOOS; Rintoul et al., 2012; Newman et al., 2019). In the first Antarctic and Southern Ocean Horizon Scan of SCAR (Kennicutt et al., 2014), the scientific community identified the need for a better understanding of systems, which can only be achieved by targeted long-term observations, measurements and analyses. The long-term IEAMaR work can significantly build upon the recently ended SCAR biology programs "Antarctic Thresholds - Ecosystem Resilience and Adaptation" (AnT-ERA; Gutt et al., 2013c) and "State of the Antarctic Ecosystem" (Ant-ECO), as well as the ongoing program Ant-ICON, and benefit from their networks of communication between experts. Moreover, the IEAMaR initiative shall underpin the efforts within CCAMLR to establish the WSMPA (Teschke et al., 2020b), by providing key reference sites to establish the required WSMPA research and monitoring plan, and enrich various national research programs in the wider IEAMaR region. All data acquired during the IEAMaR work shall be stored in international data repositories with general scope or maintaining specific information for standard analyses. For example, surface $pCO_2$ data shall be submitted to the annual updates of the Surface Ocean

CO$_2$ Atlas (SOCAT) (Bakker et al., 2016), which is widely used for studies on regional and global scale. SOCAT,
in turn, informs the Global Carbon Project for its annual update of the Global Carbon Budget. Biogeographic and
biological trait data shall be made available through the "SCAR Antarctic Biodiversity Portal"
(https://www.biodiversity.aq, last access: 23 August 2022), whilst a broad variety of other ecological data shall
be published in the "PANGAEA Data Publisher for Earth and Environmental Science" (https://www.pangaea.de,
last access: 23 August 2022). Genetic and biodiversity data can further be stored in the "Barcode of Life Data
System" (https://www.boldsystems.org, last access: 23 August 2022).
The Southern Ocean and Antarctic continent are managed within the framework of the *Antarctic Treaty System*,
which is based upon scientific understanding and environmental protection. Some of the societal needs and
challenges may overlap with a global context while others are and will remain unique (Van de Putte et al., 2021).
In general, all information, including data and their interpretation shall contribute to international scientific
assessment programs, such as IPCC and IPBES, and other advisory bodies (Fig. 3).
**4.3 Synergies**
The long-term observational IEAMaR work will provide a unique opportunity to collect a comprehensive set of
physical, geochemical and biological key data, eEOVs and EBVs, from all three main marine ecosystem
compartments (sea ice, water column, and sea floor) on a regular basis. It shall employ a highly cross-disciplinary
approach, integrating various research fields to gather physical-chemical information about marine environments,
their exchange with other Earth system compartments, and investigate biological and ecological processes over a
wide range of scales, from biomolecules to organisms to ecosystems, and from weeks to decades (Constable et
al., 2016; Gutt et al., 2018). Importantly, the integration of the research needed for the proposed WSMPA and the
long-term IEAMaR concept shall bring fisheries scientists and marine ecologists together, with experts from all
the CCAMLR member states, to explore the benefits of cross-science approaches and international collaboration
(Teschke et al., 2020b).
Long-term observational and other scientific work in the IEAMaR region, being representative of the Antarctic
Coastal Current (ACoC) in combination with the Weddell Gyre, will benefit from the fact that this area has already
been sampled for decades. For example, Fimbul is the southernmost part of the long-term hydrographic repeat
section along the Prime Meridian between South Africa and Antarctica (e.g., van Heuven et al., 2011), and the
eastern part of a hydrographic transect through the entire WS that starts off the Kapp Norvegia (Fig. 1; Strass et
al., 2020). A novelty, and an added value, of the IEAMaR framework shall be the establishment of a coordinated
and integrated ecological program in the eastern SO, applying highly standardized protocols for the sampling of
material, analyses and observations to make the results directly comparable with comparative studies over space
and time. On a wider geographic scale, the long-term IEAMaR observatory in the eastern WS and DML coast
shall be integrated with similar research performed in the Filchner-Ronne region in the southernmost WS (e.g.,
Hellmer et al., 2012; Daae et al., 2020), and with the above-mentioned three hydrographic transects in their entire
lengths far to the north and west, respectively. Moreover, comparisons shall be possible between the currently still
relatively stable East Antarctic IEAMaR sub-areas with the already drastically changing regions east and west of
the Antarctic Peninsula (e.g., Lin et al., 2021).
The concept presented herein shall also be well suited to raise the awareness of the public, including school classes,
for a healthy marine biosphere. Moreover, it shall provide perfect opportunities for education and training of a
future generation of polar researchers through generating unique occasions for joint cross-disciplinary data
analyses and thematically targeted fieldwork, which provides results being highly relevant to society (Kennicutt
et al., 2014; Xavier et al., 2019).
**5 Conclusions**
A major conclusion from global and regional assessments is that the detection of the impacts of climate change
on ecosystems demand long-term ecological observations and an improved understanding of ecosystem
functioning and its drivers (Rogers et al., 2020). Such studies can also provide insights into ecological processes
in an applied context, e.g., climate-driven modifications of ecosystem services such as oxygen production and
biological $CO_2$ uptake or potential changes resulting from other anthropogenic impacts. Closing knowledge gaps
in this context would provide a sound and independent basis for the current discussion - especially on a global
reduction of greenhouse gases, since transformation strategies, proposed for intensively used ecosystems and
nature-based solutions, are hardly options for the Antarctic. Such studies can also provide valuable information
on the effectiveness of the proposed WSMPA. The pressure from stakeholders to address such unanswered applied
ecological questions should foster coordinated cross-disciplinary and international research, in which major
advances are to be expected in more than single disciplines. The envisaged framework for long-term studies in
the IEAMaR region will also increase our knowledge about first-principle issues, e.g., on energy flow in food
webs and on biodiversity patterns including their dynamics. The collected data are to be made publicly available
for policy makers to facilitate appropriate actions and recommendations. Furthermore, the expected findings of
IEAMaR studies shall be suitable for publication in textbooks and in public media.
The stand-alone feature of the IEAMaR concept lies in the particularly extensive integration of long-term physical,
geochemical and biological research, which allows for gaining unique ecological insights. The profoundly
enhanced system understanding will provide evidence for temporal variabilities of both environment and
biodiversity, which can be attributed with high confidence to ongoing climate change or variability. The data will
also feed into ecological projections in response to anthropogenic climate change, as well as fishing pressure.
Both kind of results are urgently demanded by forthcoming IPCC and IPBES reports and address some of the
aims of the UN Sustainable Development Goals, especially #13 "Take urgent action to combat climate change
and its impacts" and #14 "Conserve and sustainably use the oceans, seas and marine resources for sustainable
development". These initiatives and other assessments have the final aim to contribute through a healthy
environment to the wellbeing of humans, also in remote large areas such as the SO.
**Author contribution**
JG, DP and FM contributed most to develop the concept, wrote the general text incl. conclusions and contributed
to themes 1-3. HG and AVdP contributed to the general concept and text, H-OP to the international
implementation, SA, OE, CHa, MH, TH, EI, MJ, SM, and STr mainly to theme 3.1, DKAB, CHe, HB, HF, CLB,
HL, FM, FS, IvO, MW, and DZ to theme 3.2, TB, SH, SM, KT, and ST to theme 3.3. All authors finalized the
entire text document.
**Competing interests**
The authors declare that they have no conflict of interest. Some authors are guest editors of the special SOOS-
volume. The peer-review process was guided by an independent editor, and the authors have also no other
competing interests to declare.
**Acknowledgements**
Thanks are due to Rebecca Konijnenberg, Hendrik Pehkle and Yves Nowak (all AWI) for the design of Figs 2
and 4, respectively.

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
