# Peer review of "Reviews and syntheses: A framework to observe, understand, and project ecosystem response to environmental change in the East Antarctic Southern Ocean"

_Biogeosciences, 2022_

## Referee Comment (RC1)

Review of Gutt MS

This is a long difficult to read manuscript!  It reads as though it were composed by a large committee, which of course it was.  It is long and very terse which is necessary because it covers an immense amount of ground. Simply put, it is the best LTER proposal I have ever seen.  Yet at the same time, it is embarrassing but I have tried to read it three times and really don't understand all that much of it.  Yet this is its strength – I suspect that very few people really understand all of it.  All the other LTER and MPA proposals and discussions have had a much more narrow focus, usually biology, geochemistry or fisheries and they basically ignore very important physical parameters as they focus on more narrow population or ecosystem dynamics.  Here the authors have pulled together all the physical and biological parameters that I can think of (actually more than I had ever thought of) into a coherent and unique framework.

As a reviewer I attempt to understand a manuscript, and then go through the manuscript section by section offering constructive suggestions. In this case as I go through it all I seem able to do is nod my head and agree with the assertions, hence I am not a very useful reviewer.  Perhaps the one issue that is missing has to do with fishing impacts which has had such a large but unstudied and virtually unknown ecosystem impact on the Ross Sea.  But in your case, there has not been much if any fishing so you really have nothing to talk about except for the virtual removal of whales, so this is not a criticism, but perhaps a suggestion to consider potential fishing impacts a bit more carefully.  Yet, I agree that they are minor compared to the climate change problems that you do focus on very carefully.

Your focus on long term observations is sound, yet as you point out, these always lack the physical drivers of the phenomena that the observations record.  For example, my own career was spent at McMurdo where I was not allowed to maintain the temporal continuity that I had hoped to have for the observations, but I was still able to report massive changes after decades of relative stasis.  Yet the oceanographic and climatic forces actually driving those changes are still a mystery. It is not for lack of interest as there were many proposals to study the oceanography, yet the NSF insisted on investing in very expensive if interesting lake studies and redundant seal programs. So, despite several programs looking at specific biological questions that relied on oceanographic data, those baseline or benchmark data do not exist. More generally, the Ross Sea might be the most interesting ocean system in the Antarctic, but the fishing interests and political clout of CCAMLR has focused on defending at all costs the toothfish industry rather than collecting the type of data you propose in this manuscript. It is this background that makes this paper so appealing to me.  Starting around line 117 you describe your objectives and the existing data, and while I have tried to have critical suggestions, I can offer nothing but praise for this effort.  Your "Overarching concept" is simply superb.

The method section is excellent, integrating both spatial and temporal scales with large scale processes.  And importantly, it is not a wish list fantasy because the authors focus very carefully on the actual sampling techniques.  I have no expertise in these procedures, but they seem

consistent with all the oceanographic research I know. You have done an excellent job deciding where to sample. These decisions are based on very solid understanding of the processes and are well defended and appropriate. I just assume that the data management section is adequate.

The three LTER themes are as good as any I have ever seen; indeed, they are much much better than any I have ever seen, especially the other LTER project around the united states that utterly lack the excellent scientific overviews offered in these long sections.

3.1 ecosystem drivers. This long-detailed section offers exactly what is missing in most other projects. It is exactly what we so desperately needed at McMurdo sound (and American kelp projects as well for that matter).

3.2 Ecosystem functioning components represents the focus of the other good LTER studies. The authors have integrated these other projects very well such that this proposal is solidly based on the results of the other very good data that exist. The discussions of limitations and strengths are carefully presented. These are the issues I should know about, but the authors have dug up way more literature than I know, and I can only applaud this section.

3.3 Ecosystem services in my mind represent a form of wishful thinking with the fantasy that fishing organizations such as CCAMLR will respect the data, but it is a necessary section and is exceptionally well thought out and presented. Many of the objectives such as defining the carbon sink, developing biochemical processes, protection of rare species, and environmental processes are critical to all aspects of the future management. While past political winds certainly have made me hyper cynical, the presentation here is powerful and based on very solid scientific thinking. Amazingly, I almost find myself optimistic that a program such as this can be funded. If it can be funded, it will set a powerful example for the rest of the world.

I hope other reviewers can offer more constructive suggestions, since I find myself applauding the manuscript as it is!

I am worried about the prospects of implementing this excellent proposal but accepting this manuscript and publishing it is the first step. Then all of us will have to focus on the difficult political battles to implement it as written rather than have it manipulated to serve special interests. Obviously implementing this program will depend on a new political vehicle that is not crippled by a consensus rule! But first we need to get this manuscript published and read.

---

## Referee Comment (RC2)

This is a lengthy and detailed paper describing a potential multidisciplinary long term observatory in the eastern Weddell Sea close to the Antarctic coast along the Dronning Maud Land. Eberhard Fahrbach would be proud of this legacy. I enjoyed the bringing together of all the disciplines. The paper is a useful concept and is publishable, although I think it can and should be further strengthened.

**Major or general comments:**

1. The whole paper focuses a bit too much on what we know and not enough on what we *don't* know.
2. I think it would be helpful to be more consistent with present day literature, in distinguishing between the Antarctic Coastal Current (along the coast or ice shelf front, on the continental shelf) and the Antarctic Slope Current (along the slope). I appreciate that many of us have not been careful in our usage and in the proposed region these often merge. But this is not the case elsewhere round Antarctica, and it might lead to confusion if these phenomena are not defined carefully and distinguished appropriately.
3. There is no mention of the Antarctic Slope Undercurrent. Personally I think this is an important (though small) current, that can play a role in eastward transport of nutrients, trace elements, larvae, biota, etc, beneath the westward-flowing Slope Current. The proposed region would provide an excellent opportunity to study this role (see for example Chavanne et al. (2010) at 18W).
4. I found it frustrating that the paper doesn't actually say where the survey sites should be, or how many sections/stations there should be. Everything is "to be determined". It is tempting to say that it would be better for the authors to propose locations for surveys before they publish this. Otherwise what is the use of the paper?
5. All places that are referred to anywhere in the text should be labelled on a map. For example Gunnerus Ridge and Atka Bay.
6. I found the distinction between LTO and LTER unclear. Since it's clearly based around Neumeyer, I wondered why you weren't calling it the Neumeyer LTER analogous with Palmer LTER? For me at least, I know where Neumeyer is roughly, whereas I'd struggle with the unpronounceable acronym and where it refers to?
7. There is mention of fisheries; how are you going to monitor fish populations, age distributions, species, etc?

**Comments by section:**

**Section 1 Introduction**

I found the Introduction quite unfocussed. I recommend shortening this section and thinning out the references considerably. Many of the paragraphs read like a random selection of papers and results from different places round Antarctica and it was unclear why they were relevant to this paper? For example, we are told about mortality of sponges in McMurdo Sound. All very interesting, but why is this relevant? The link to this paper, or lessons learnt, need to be stated. For example it might be "we were able to learn about the mass mortality of sponges because someone made measurements every year", or whatever.

Perhaps it would be better to think of each paragraph as conveying a key message or answering a specific question. For example, the Introduction could have 3 paragraphs: (1) what is the international context for the proposed LTO/LTER? (2) what can we learn about good observing system design from the other LTOs/LTERs? (3) what is the background of historical research in the proposed LTO/LTER region?

The first two paragraphs of the paper are really just a list of a large number of previous reports and programmes, which is quite dull to read and doesn't give a sense of excitement. It would be good if this could perhaps be shortened to "what the reader needs to know for what comes later"?

The next paragraph (L71-80) gave a series of sentences that seemed disconnected; it was unclear what the message of the paragraph was.  What is the relevance to this paper of changes seen at the WAP?

The paragraph about the observatory (L117-138) seemed out of place?   Should you not explain why it is needed first, and this comes later?  I would shorten this paragraph considerably and ensure this information comes in later where appropriate.

**Section 2 Overarching concept**

I found the statements of knowledge gaps underwhelming; if I received this as a grant proposal I would find it difficult to support it.  It would be good if this section could be strengthened since it is surely a key contribution of the paper?   Many of the points made are vague.

For example, what do you mean by "long-term" in (1)?  Why are these data lacking, and what do we have now to alleviate this lack that we did not have before?  Maybe these data were not collected before because it's not important to do so?  (ok I know that is not the case, but you don't say why they are important).  Maybe we only now realise the importance?  Maybe only now do we have the tools?

Point (2) is rather aggressive?  If these protocols are often ignored, why will establishing an LTER help?  I wonder if this could be phrased more positively?

**Section 3**

Section 3.1.2 in particular talks about "lack of understanding" but all we have been told about is what we do know. We've not been informed what we lack understanding of.  The section about biology (3.2.1) is better in this respect, identifying key knowledge gaps. Section 3.2.1 needs to be refocused on clarifying what these knowledge gaps are.

Somewhere in the paper, I'd like to have seen a brief critical assessment of what design knowledge can be transferred from the Palmer LTER to the system design of this LTER. E.g. what horizontal, vertical and temporal resolutions are needed. Are there lessons that can be learnt about what worked and what didn't?

Could you add more justification for three sites? Why 3? Why not 2 or 4?  You argue that the conditions in the region are typical of the East Antarctic coast, but this argument is undermined by suggesting three sites.  What do you learn (or gain) by having more than one?

The methodology section 3.1.3 has oddly almost no references and yet this is where I would have found these useful to envisage what you have in mind.

Schematics or diagrams of the proposed observing system would be very useful. E.g. what depths will you monitor? Where might moorings be? What water depth?  Moorings are great but presumably you won't risk sensors in the upper few hundred metres because of the iceberg snag risk. How will you monitor the upper ocean critical for biology?  I'd have liked more information about technologies envisaged.

I agree that a mooring through Ekstrom Ice shelf would be great, but you need to articulate why. If you're only interested in shelf water properties as you state, you could instead make really comprehensive mooring arrays at the entrance and exit of the ice shelf for the same cost and effort as one hot water drilled hole.

I found no mention of bathymetry? Key for all disciplines. Is the bathymetry sufficiently well known? Has the whole region had multibeam surveys?

There was no mention of meteorology? Is there the local met monitoring and might it need supplementing?

No mention of ice shelf monitoring? What about ApRES on ice shelves, as meltwater is important for e.g. iron and stratification?

Section 3.3.3, how will you monitor surface pCO2 in 'a coastal region' hourly year-round? This will have major logistical challenges, e.g. sea ice and ice bergs, and a single location might not be representative of the region. How will you choose the location? This methods section would benefit from more detail; it is very short.  Add references.

How will you ensure you can obtain winter time measurements? Does it matter if the CTD sections are occupied at different times of year each year? How will you ensure study of the whole plankton bloom?

**Section 4**

Section 4.1.  It would be useful to state how the data will be made available – storage in a data centre is not the same as that data centre making the data freely and openly available.  You might identify which data will be made available in (near) real time, and which will be available are an embargo period.

Section 4.2 states that a comprehensive data set will be obtained from sea ice, water column and sea floor.  I may have missed it, but I don't recall discussion of measurements of the ecosystem of sea ice?  I only read about measurements of the ecosystem beneath sea ice, from ice stations?  How will you be accessing ice stations?  How will you choose locations for sea ice stations?  Will they be on fast ice or ice floes?  How frequently will they be monitored? Throughout the year? More explanation is needed (probably in section 3).

Section 4.3 would benefit from references.

**Minor comments:**

L34 I wouldn't call it easterly – not clear what is meant here – omit?

L37 I didn't understand what was meant by "and sound data can act as a model to develop and calibrate projections".  What do you mean by sound?  How can data act as anything?  How can a model develop or calibrate a projection?

L108-110 Repetition of HAFOS, rephrase sentence.

L125 can you give an idea of how many "a number" will be?

L126 what does sensu mean?

L182- 183 You have east and west round the wrong way – the ASC flows west and the ACC flows east.

Numbering of section 3.3.2 is incorrect, p23

L710 I wasn't sure what this sentence meant.  Grammatically the first line (l710) didn't seem to make sense.

**Figures:**

**Fig.1**  This is a nice introductory figure.  However it is not much described.   To be useful to readers, especially those who do not know the currents and water masses, this figure needs

further explanation.  The relevance of the water masses and circulation to the ecology/biogeochemistry should be stated.

**Fig. 2**. Label for Kapp Norwegia is needed.  The caption is confusing because it says there are three areas but the highlighted region is all one?  Are the three labels meant to be on top of the survey sites?  I'd like to see a map where the boundaries and properties for the three regions are clearly shown.  It might be helpful to have a map of the three regions showing sea ice minima and maxima, SST, or something similar?

**Fig 3 -** check that the acronym soup is already defined, and define in caption for maximum readability.

I couldn't find where figure 4 is discussed?

Karen J. Heywood
University of East Anglia

---

## Author Comment (AC1)

Dear Carol Robinson,

Please find below our comprehensive responses to all the criticisms by the reviewers K. Heywood and Paul Dayton. The basis for these responses is a thorough revision of the manuscript, which we can resubmit immediately in the case that you accept the changes announced in the responses below.

Kind regards, also on behalf of all co-authors
Julian Gutt,

**Review of Gutt et al., submitted to Biogeosciences Discussions, Karen Heywood (and Paul Dayton)**

**K. HEYWOOD:** This is a lengthy and detailed paper describing a potential multidisciplinary long term observatory in the eastern Weddell Sea close to the Antarctic coast along the Dronning Maud Land. Eberhard Fahrbach would be proud of this legacy. I enjoyed the bringing together of all the disciplines. The paper is a useful concept and is publishable, although I think it can and should be further strengthened.

*Reply: We appreciate the constructive comments and suggestions. We addressed all of them in appropriate modifications of the manuscript, and we acknowledge that the manuscript as a whole has been significantly strengthened through the revision, particularly by making the main objective clearer. In the following we respond explicitly to all points, including the second referee's (**P. Dayton**) only concrete recommendation (fisheries). In some cases, we provided comprehensive additional information with our response. However, as the manuscript was already considered to be "lengthy", we generally tried to address the comments and suggestions by changes that may be quite significant in terms of content at times but do not lead to a lengthening of the text.*

**Major or general comments:**

**K. HEYWOOD:** 1. The whole paper focuses a bit too much on what we know and not enough on what we *don't* know.

*REPLY: We agree that the knowledge gaps, which are important to derive the objectives, have been addressed too late in the current ms. Therefore, we moved them to a more prominent place in the introduction. Moreover, we moved the somewhat less significant (but in our opinion also necessary) paragraph about "lessons learned from previous long-term studies" backwards to chapter 4. In the revised introduction, just two sentences explain that some long-term observatories already exist. Finally, we phrased the general knowledge gaps (in the introduction) more concretely by including key terms such as driver quantification, system understanding, biodiversity shifts/heterogeneity and end-to-end observations/simulations. See also below our response to the first comment on Section 1.*

**K. HEYWOOD:** 2. I think it would be helpful to be more consistent with present day literature, in distinguishing between the Antarctic Coastal Current (along the coast or ice

shelf front, on the continental shelf) and the Antarctic Slope Current (along the slope). I appreciate that many of us have not been careful in our usage and in the proposed region these often merge. But this is not the case elsewhere round Antarctica, and it might lead to confusion if these phenomena are not defined carefully and distinguished appropriately.

*REPLY: We agree that the ACoC and ASC are two separate circulation features. Accordingly, we changed the text to: "Large-scale oceanographic features exposed to climate change impact this region (Fig. 1): The Weddell Gyre branches off the eastward flowing Antarctic Circumpolar Current (ACC; van Heuven et al., 2011) and converges between Gunnerus Ridge (30 °E) and the Ekström Ice Shelf (8 °W) with the westward flowing Antarctic Coastal Current (ACoC) near the coast and ice shelf fronts and the Antarctic Slope Current (ASC) along the continental slope facilitating zonal connectivity and shaping the coastal environment."*

**K. HEYWOOD:** 3. There is no mention of the Antarctic Slope Undercurrent. Personally I think this is an important (though small) current, that can play a role in eastward transport of nutrients, trace elements, larvae, biota, etc, beneath the westward-flowing Slope Current. The proposed region would provide an excellent opportunity to study this role (see for example Chavanne et al. (2010) at 18W).

*REPLY: We acknowledge that the significance of the undercurrent was not adequately addressed in the ms and added a sentence on this issue (as proposed): "A quantification of the Antarctic Slope Undercurrent will help to determine its role in eastward transport of nutrients, trace elements, larvae, biotas, etc beneath the westward-flowing Slope Current."*

**K. HEYWOOD:** 4. I found it frustrating that the paper doesn't actually say where the survey sites should be, or how many sections/stations there should be. Everything is "to be determined". It is tempting to say that it would be better for the authors to propose locations for surveys before they publish this. Otherwise what is the use of the paper?

*REPLY: First, we'd like to point out that this paper is neither meant as a grant proposal nor a detailed science plan. Its main purpose, which we explain more clearly in the revised ms in response to the criticisms, is to provide a framework, based on sound scientific evidence available to date, for such concrete project applications, which we hope will be developed in the future. To clarify this basic approach, we included point 2 in 1.3 Objectives: "Lay out a conceptual framework for upcoming work, time and cost plans as well as a thorough sampling design", in addition, we avoided the terms "proposal" or "proposed" throughout the revised ms.*

*Moreover, we hope that through the revision it is more unambiguous than before that our main intention is to present a deliberately broad framework or concept to the public (via Biogeosciences) for a wider discussion, based on evidence-based arguments about why a long-term observatory in the East Antarctic Southern Ocean is urgently needed, what major information it should deliver, where it should therefore be placed, and how a general field sampling scheme could look like (see also our response to the last comment of Section 1 below), We are convinced that such a framework will provide an important basis for any further steps towards developing a research project and implementation plan. In our opinion, it would be counterproductive, however, if we would provide in our concept paper too many details on, e.g., finances, personnel, equipment, ship time etc., because it*

*would narrow the options for a later grant proposal (which could refer to our concept paper to pinpoint its motivation and approach).*

*However, we also acknowledge the valid criticisms that in the submitted ms the description of the concept was generally too vague and we addressed the request for some more details about the possible later implementation of the presented framework, by revising the ms accordingly at various places. For instance, the term "to be determined", which appeared twice in the original manuscript, has been changed/deleted. Moreover, we clarified in the revised ms that the three circles in Fig. 2a represent the three regions, which we suggest as potential study areas best suited to address the main objectives, and we include, as an example, a schematic design (Fig. 2b), how (mostly) within these areas the stations can be arranged. Furthermore, we provide, as recommended, more details about the depths of transects to be sampled in 2.3: "Shipboard and autonomous data collections are suggested to take place at water depths ranging from the coastal shelf, including unusually shallow sites like the Norsel Bank off Kapp Norvegia (approx. 60 m), the slope between approx. 450 and 3000 m depth, influenced by the ACoC, to the deep sea, influenced by the Weddell Gyre. Moreover, historical transects, such as the Prime Meridian, should be extended (Fig. 1)." For further revisions in this regard, see our responses to the comments on the specific sections of the ms below.*

[Figure]

[Figure]

*For legend see below.*

**K. HEYWOOD:** 5. All places that are referred to anywhere in the text should be labelled on a map. For example Gunnerus Ridge and Atka Bay.

*REPLY: In the revised ms, all names of places referred to in the text (e.g., Gunnerus Ridge, Atka Bay, Norsel Bank and Kapp Norvegia) are included in Fig. 1 or Fig. 2b.*

**K. HEYWOOD:** 6. I found the distinction between LTO and LTER unclear. Since it's clearly based around Neumeyer, I wondered why you weren't calling it the Neumeyer LTER analogous with Palmer LTER? For me at least, I know where Neumeyer is roughly, whereas I'd struggle with the unpronounceable acronym and where it refers to?

*REPLY: We included a sentence in 1.3 Objectives: "The long-term observatory represents the location and logistic infrastructure of the intended observational work, while LTER refers to the scientific studies to be carried out there". In addition, we just use the terms "long-term observatory" or "observatory" throughout the revised ms but do not use the acronym LTO anymore. Lastly, we understand the recommendation to avoid an "unpronounceable acronym" (such as "WSoDML") to denote the study area and the intended observatory itself. To find a new name and acronym triggered a discussion among the authors. We have to be content and polically correct. As a consequence, we propose now: "Integrated East Antarctic Marine Observatory" (IEAMO).*

**K. HEYWOOD:** 7. There is mention of fisheries; how are you going to monitor fish populations, age distributions, species, etc? Se. also review by **P. DAYTON:** Perhaps the one issue that is missing has to do with fishing impacts which has had such a large but unstudied and virtually unknown ecosystem impact on the Ross Sea.

***REPLY:*** *In the revised ms**,** we added text on fish and fisheries by including/extending these two paragraphs in 3.3.2 and 3.3.3:*

*"LTER work at IEAMO shall also provide insight into the biology, life cycle and trophic role of Antarctic toothfish. Since 2014, every year around 200 tonnes of Antarctic toothfish are caught by fishing vessels operating in the CCAMLR research block 48.6_5 in front of the DML coast (Fig. 5). Combining the information obtained by these longline operations with data obtained at the IEAMO will allow comparisons of the benthic habitats in this research block with similar habitats in unfished areas to study the physical disturbance and effects of longline fishing. The close proximity and partial overlap of the IEAMO region with the CCAMLR research block provides also the opportunity to carry out further studies of local and regional food web and ecosystem effects caused by the annual removal of large quantities of D. mawsoni as a top demersal predator. Less consumption will certainly have impacts on its prey species and the entire seabed community. In addition to this predation release effect, changes to toothfish abundances caused by fisheries could also impact toothfish predators like Weddell seals and whales (sperm, killer and Arnoux beaked whale)."*

*"In addition, scientific sampling of the benthic and pelagic fish fauna should be carried out in areas designated for this purpose. These studies will be designed to close gaps and complement on the one hand the results of historical fish fauna research carried out in the 1980s and 1990s e.g. by the Alfred Wegener Institute, and on the other hand the data on toothfish, toothfish prey and by-catch species obtained in the commercial long-line operations in the CCAMLR fisheries research block 48.6_5 (see Fig. 4). This will contribute to both the development of a stock hypothesis of Antarctic toothfish in the larger Weddell Sea area and to the research and monitoring efforts described in the Weddell Sea MPA proposal."*

**Comments by section:**

**Section 1 Introduction**

**K. HEYWOOD:** I found the Introduction quite unfocussed. I recommend shortening this section and thinning out the references considerably. Many of the paragraphs read like a random selection of papers and results from different places round Antarctica and it was unclear why they were relevant to this paper? For example, we are told about mortality of sponges in McMurdo Sound. All very interesting, but why is this relevant? The link to this paper, or lessons learnt, need to be stated. For example it might be "we were able to learn about the mass mortality of sponges because someone made measurements every year", or whatever.
Perhaps it would be better to think of each paragraph as conveying a key message or answering a specific question. For example, the Introduction could have 3 paragraphs: (1) what is the international context for the proposed LTO/LTER? (2) what can we learn about good observing system design from the other LTOs/LTERs? (3) what is the background of historical research in the proposed LTO/LTER region?
The first two paragraphs of the paper are really just a list of a large number of previous reports and programmes, which is quite dull to read and doesn't give a sense of excitement. It would be good if this could perhaps be shortened to "what the reader needs to know for what comes later"?

*REPLY: We acknowledge that the organization of introduction needed improvement to more plausibly develop background, motivation, basic approach and major objectives of our concept paper. In response to K. Heywood's comments, we deeply revised the Introduction in various ways. First, we added subheadings to clarify the organization of the introduction and ease reading: (1.1) Background, (1.2) Knowledge gaps, and (1.3) Objectives. The revised section 1.2 contains text moved forward from section 2 ("Overarching concept") to the "Introduction", to make the derivation of the objectives in section 1.3 more plausible.*

*Text on existing LTERs has been moved backwards to section 4.1 ("Lessons learned"). Such LTERs are mentioned in the revised introduction only with two sentences (see also response on the following comment below). In the more detailed information in section 4.1, we checked all previous LTERs for their clear and relevant message in the context of our paper and changed text accordingly (e.g., on sponge mortality, as proposed, as well as on emperor penguin habitats and foraging behaviour). However, we did not include additional details because K. Heywood criticised already (see above "Major comments (1)") the revision character of this paragraph (note, however, that our manuscript falls into the category of a "revision", and, thus, providing such background information seems to be generally justified). The general international framework (regarding the relation of our concept to IPCC, IPBES, SCAR, etc.) is mentioned in the 1st paragraph of the introduction, more details in "4.2 International integration".*

*Finally, we acknowledge that the number of references was excessive, and we decreased this number by approx. 18 in the revised ms. However, we'd like to point out that due to its transdisciplinary scope, a relatively high number of references seems generally justified Moreover, please note that in response to some of the comments we also had to include a few additional references.*

**K. HEYWOOD:** The next paragraph (L71-80) gave a series of sentences that seemed disconnected; it was unclear what the message of the paragraph was. What is the relevance to this paper of changes seen at the WAP?

*REPLY: We agree that in the introduction of our paper such detailed information was misplaced, as it distracted from the main message. However, it is surely necessary to mention previous and existing LTERs and what we can learn from their findings. Thus, such information has not been completely deleted but moved backwards to section 4.1 (see above). The differences between WAP and East Antarctic regions are made clearer, and a circumpolar study is now also mentioned in the revised text.*

**K. HEYWOOD:** The paragraph about the observatory (L117-138) seemed out of place? Should you not explain why it is needed first, and this comes later? I would shorten this paragraph considerably and ensure this information comes in later where appropriate.

*REPLY: Adding the subheading "1.3 Objectives" to the Introduction (see above) should help to understand that this paragraph deals with the major aims of the manuscript, placed as usual at the end of the introduction of scientific papers. The "Why it is needed" is now addressed in the preceding section 1.2 ("Knowledge gaps") of the Introduction, moved upwards upon the suggestion from former section 2 (see also our response abov). However, to make its message more focussed and clearer the paragraph has been shortened by moving text about eEOVs and EBV to section 2.3 ("Methodological*

*approach"). To reduce the complexity of the manuscript, avoid redundant text and, thus, make the objectives even clearer we included the content of former section "2.2 Scientific goals" in the revised section "1.3 Objectives". See also our response to the Major Comment 4 above.*

**Section 2 Overarching concept**

**K. HEYWOOD:** I found the statements of knowledge gaps underwhelming; if I received this as a grant proposal I would find it difficult to support it. It would be good if this section could be strengthened since it is surely a key contribution of the paper? Many of the points made are vague.

*REPLY: This comment does largely overlap with the previous comments that we responded to above. We'd like to point out again that this paper is not (and should not be confused with) a project proposal. Instead, we strive for presenting a concept that can hopefully provide a framework for a later grant application. Consequently, our manuscript is - on purpose - not a science or work plan for the implementation of the presented framework. To emphasize this approach more clearly, we deleted the terms "proposal" or "proposed" in the text where necessary. However, to address the general comment on the vagueness of our description, we illustrate in the revised ms where the subareas of interest could be placed and how transects and stations could be placed in such areas but **only as a scheme** (Fig. 2b). We also included, as proposed, information on the depth range to be covered in the observational work (see response to the general comment 4 above). Furthermore,*

*(1) Text about knowledge gaps have been moved upwards to a more prominent place in the introduction*

*(2) Knowledge gaps are addressed with a certain granularity in five separate points. We argue that more details are not adequate, since our manuscript focuses on "better" future research rather than deficits of past research. We feel that this approach is in line with the recommendation not to emphasize too much the ignorance of protocols in the past (and we made this statement a bit more vague).*

*(3) At the beginning of the revised section 1.2, we list now a few quite general but comprehensive knowledge gaps: "… the current scientific knowledge in terms of a quantification of physical-chemical ecosystem drivers, an understanding of ecosystem processes and of temporal shifts of biodiversity as well as its spatial heterogeneity, is insufficient".*

**K. HEYWOOD:** For example, what do you mean by "long-term" in (1)? Why are these data lacking, and what do we have now to alleviate this lack that we did not have before? Maybe these data were not collected before because it's not important to do so? (ok I know that is not the case, but you don't say why they are important). Maybe we only now realise the importance? Maybe only now do we have the tools?

*REPLY: In the revised ms, the term "long-term" is now explicitly defined (e.g., in the Abstract: "Here we present a framework for establishing a long-term cross-disciplinary study on decadal time scales…"; in the Objectives: "Regular observational work should be conducted over a period of decades..."; and in section 2.1: "...on a decadal time scale".). In section 1.2 we write "understanding of ecosystem processes … is insufficient for a*

*number of reasons", and those reasons are mentioned in the following text. We state that "gaps of knowledge and knowledge transfer have not yet gained sufficient acceptance". However, we would not like to analyze in detail the reasons why data are lacking (and thus 'blame' researchers, decision makers or funding agencies). In addition, we added: "Thirdly, advanced tools were not available and important background information did not exist in the past." We think the revised "knowledge gap" section 1.2 is the best place to explain this and it seems not necessary to repeat this in chapter 2.*

**K. HEYWOOD:** Point (2) is rather aggressive? If these protocols are often ignored, why will establishing an LTER help? I wonder if this could be phrased more positively?

***REPLY:*** *Text revised accordingly: "… standardized protocols … have not frequently and consequently been implemented". Establishing a LTER will not only help to solve the problem but also to increase the awareness of the need of such long-term observatories, better highlighted now in the revised section 1.3 ("Objectives").*

**Section 3**

**K. HEYWOOD:** Section 3.1.2 in particular talks about "lack of understanding" but all we have been told about is what we do know. We've not been informed what we lack understanding of. The section about biology (3.2.1) is better in this respect, identifying key knowledge gaps. Section 3.2.1 needs to be refocused on clarifying what these knowledge gaps are.

***REPLY***: *Section 3.1.1 is meant to describe the status quo of knowledge, with both its knowns and some unknowns (some of which have been added to the revised manuscript). The following section 3.1.2 then provides the list of objectives, including the lack of understanding associated with every single objective. After the list of objectives some more info on the lack of understanding and the lack of data are presented. Accordingly, this section starts with: "To address the current lack of understanding of observed and expected changes in the physical-chemical environment and their impacts on biogeochemical fluxes in the marine ecosystem, we shall address the following objectives (for closely related ecosystem services see section 3.3):…"*

**K. HEYWOOD:** Somewhere in the paper, I'd like to have seen a brief critical assessment of what design knowledge can be transferred from the Palmer LTER to the system design of this LTER. E.g. what horizontal, vertical and temporal resolutions are needed. Are there lessons that can be learnt about what worked and what didn't?

***REPLY:*** *The environmental conditions in the area of the Palmer LTER differs considerably from those in the IEAMO area. As a consequence, a paragraph was added to section 4.1. "Lessons learned": "However, the pelagic ecosystem of the western Antarctic Peninsula is different in that the shelf extends much further than in the East Antarctic investigation area. Therefore, it offers more iron sources and leads to an overall more productive ecosystem. In addition, the coastline is much more irregular and provides a special habitat heterogeneity. The Palmer LTER was established in this area in 1990 (Smith et al., 2013), at a time when climate was already changing rapidly in this Antarctic region. Surveys identified changes in pelagic food webs west of the Antarctic Peninsula (Ducklow et al., 2006), where benthic inshore biodiversity has partly increased as a result of long-term glacier retreat at King George Island (Sahade et al., 2015; Zwerschke et al., 2022). However, the trends detected*

*in primary production and higher trophic levels are inconsistent, largely due to the heterogeneity in sea-ice dynamics that in turn depend on variable meteorological conditions (e.g., Montes-Hugo et al., 2009; Lin et al., 2021), with consequences for oceanic $CO_2$ uptake (Brown et al., 2019). The early establishment of the Palmer LTER and additional studies covering the area from the northern tip to the southern base of the Antarctic Peninsula allowed researchers to detect the impacts of climate change on various marine ecosystem, highlighting the importance of establishing the "Integrated East Antarctic Marine Observatory" as early as possible, before the onset of climate-change effects in the East Antarctic SO."*

**K. HEYWOOD:** Could you add more justification for three sites? Why 3? Why not 2 or 4? You argue that the conditions in the region are typical of the East Antarctic coast, but this argument is undermined by suggesting three sites. What do you learn (or gain) by having more than one?

***REPLY***: *A sentence is included in 2.3 Methodological approach… (but not in section 3): "The concept envisages up to three sub-areas, since these should cover the different physical-chemical prerequisites and the biological heterogeneity within the wider study area, being partly representative for the East Antarctic SO (for overview see section 2.1, for details see information provided by the three scientific themes in sections 3)."*

**K. HEYWOOD:** The methodology section 3.1.3 has oddly almost no references and yet this is where I would have found these useful to envisage what you have in mind.

***REPLY:*** *Several references have been added.*

**K. HEYWOOD:** Schematics or diagrams of the proposed observing system would be very useful. E.g. what depths will you monitor? Where might moorings be? What water depth? Moorings are great but presumably you won't risk sensors in the upper few hundred metres because of the iceberg snag risk. How will you monitor the upper ocean critical for biology? I'd have liked more information about technologies envisaged.

***REPLY:*** *There is a partly new paragraph on autonomous platforms incl. moorings in 2.3 Methodological approach…: "The shipboard work would complement the higher-resolution observations performed by autonomous platforms at selected core stations, to be placed at the centres of the long-term observation transects or at existing long-term observation transects (Weddell Sea/Kapp Norvegia, Prime Meridian), to allow for the technical maintenance of the autonomous platforms and contribute to the ground-truthing of remote-sensing and modelling studies. The platforms can include various systems, such as moorings, profilers, saildrones, sea-ice buoys, gliders, benthic landers, underwater fish observatories, and time-lapse cameras, with the potential to grow into a network of autonomous observation devices."*

*Core stations are now schematically shown in Fig. 2b.*

*Several sentence referring to moorings and their risks are modified and included in 3.1.3: "The collection of oceanographic data by a mooring below the Ekstroem Ice shelf since 2005 should be continued and extended. It is well protected from the regular ice-berg traffic in front of the shelf (Oetting et al., 2022) and safer than those hung from the ice-shelf edge, which is subject to regular calving events. The mooring holds a passive acoustic*

*recorder and a CTD and has been operational until February 2022 when the iceshelf broke off and the cables were torn. This long term data series has been extremely valuable in terms of understanding physical processes at the subsurface of the iceshelf, as well as coastal oceanographic processes including the dynamics of pelagic species composition on a year-round basis and in relation to the environment. Due to the service work carried out by the overwinteres of the Neumayer station III it proves the potential longevity of such set ups - compensating for the cost of installation - which could never have been achieved by moorings deployed in front of the ice shelf where there is heavy iceberg traffic.*

*A paragraph referring to the upper ocean shaping pelagic life has been added to 3.2.3: "Autonomous bio-environmental observatories shall constitute an important pillar of LTER at the IEAMO. By combining multiple sensors on bottom-moored, sea ice-moored and free-drifting platforms, the spatio-temporal gaps between field campaigns will be closed with high-resolution data. These systems should merge existing state-of-the-art environmental sensors, such as CTDs, nutrient sensors, fluorometers, spectral radiometers and optical sediment traps, with the newest technology to monitor organisms beyond microbes. Examples for such sensors are camera systems, autonomous multi-frequency echosounders (e.g. Acoustic Zooplankton and Fish Profilers, Wideband Autonomous Transceiver), which are able to record and transmit data and to receive real-time manipulation of the sampling programme, as well as automatic eDNA samplers and imaging profilers (e.g. Underwater Vision Profiler)."*

**K. HEYWOOD:** I agree that a mooring through Ekstrom Ice shelf would be great, but you need to articulate why. If you're only interested in shelf water properties as you state, you could instead make really comprehensive mooring arrays at the entrance and exit of the ice shelf for the same cost and effort as one hot water drilled hole.

***REPLY:*** *A paragraph has been added to 3.1.3, which, however, refers mainly to the methodology (see also above): "The collection of oceanographic data by a mooring below the Ekstroem Ice shelf since 2005 should be continued and extended. It is well protected from the regular ice-berg traffic…"*

*The relevance of processes beneath the iceshelf and to study these is mentioned twice in 3.1.2 Objectives: "Understand basin-wide and climate-sensitive changes in ice-shelf/ocean interactions..." and "understand the atmosphere-sea ice-ice shelf variability and interaction, in particular drivers for pack ice and fast ice dynamics…"*

**K. Heywood:** I found no mention of bathymetry? Key for all disciplines. Is the bathymetry sufficiently well known? Has the whole region had multibeam surveys?

***REPLY:*** *Yes, bathymetry is sufficiently well known. Text is extended respectively: "As a suitable prerequisite, the bathymetry in this area is very well known, from swath sonar in the open ocean, also underneath the ice shelf from active seismic surveys (Oetting et al., 2022)."*

**K. Heywood:** There was no mention of meteorology? Is there the local met monitoring and might it need supplementing? No mention of ice shelf monitoring? What about ApRES on ice shelves, as meltwater is important for e.g. iron and stratification?

***REPLY:*** *We will use meteorological data gained at Neumayer Station III and address the need to consider ice-shelf basal melt rates in the intended analyses and included a sentence in 2.2 Methodological approach. However, we do not want to use even more acronyms, providing specific details, which are not so relevant for our general approach and which had to be explained to non-specialists. "The interdisciplinary studies at the IEAMO will benefit from meteorological routine measurements and glaciological data obtained from observations and from satellite ice shelf altimetry, including basal melt rates at the Neumayer Station III.", but we do not intend to make own meteorological measurements. This framework paper must have its limits."*

**K. Heywood:** Section 3.3.3, how will you monitor surface pCO2 in 'a coastal region' hourly year-round? This will have major logistical challenges, e.g. sea ice and ice bergs, and a single location might not be representative of the region. How will you choose the location? This methods section would benefit from more detail; it is very short. Add references.

***REPLY***: *included: "A $pCO_2$ sensor (Lai et al., 2018) will be part of a novel mooring design, which is protected against iceberg scouring and shall be deployed in a synoptic approach in combination with other instruments (see sections 3.1.3 and 3.2.3 above)."*

**K. Heywood:** How will you ensure you can obtain winter time measurements? Does it matter if the CTD sections are occupied at different times of year each year? How will you ensure study of the whole plankton bloom?

***REPLY:*** *A sentence in 2.3 Methodological approach… has been modified and extended: "These combined measurements can resolve the timing of interlinked, strongly seasonal processes and episodic extreme events by complementing ship-based snap-shot measurements with year-round high-frequency (hourly to weekly) observations of selected variables obtained through autonomous installations, such as moorings, landers and satellites (e.g., physical measurements, environmental DNA analyses and Chlorophyll-a).*

**Section 4**

**K. HEYWOOD:** Section 4.1. It would be useful to state how the data will be made available – storage in a data centre is not the same as that data centre making the data freely and openly available. You might identify which data will be made available in (near) real time, and which will be available are an embargo period.

***REPLY:*** *We strongly agree that free availability of data must be guaranteed, which is explicitly described in 2.4 Data Management. Embargo periods need to be set appropriately, however, we regard such information to be too detailed for a framework paper.*

**K. HEYWOOD:** Section 4.2 states that a comprehensive data set will be obtained from sea ice, water column and sea floor. I may have missed it, but I don't recall discussion of measurements of the ecosystem of sea ice? I only read about measurements of the ecosystem beneath sea ice, from ice stations?

***REPLY:*** *An entire paragraph on sea-ice ecology has been included in 3.2.1: "The Antarctic sea-ice itself provides the habitat for a variety of unique taxa that contribute significantly to*

*carbon flux and nutrient cycling in the SO (Monti-Birkenmeier et al., 2017; Steiner et al., 2021). Ice algae are an important source of carbon for pelagic and benthic communities (Meiners et al., 2018). Thus, the sea-ice cover critically controls ecosystem functions and services in the Weddell Sea. In addition, ice-associated biotas play an important role for the winter survival of various zooplankton taxa (Kohlbach et al., 2018, Schaafsma et al., 2017), and the sea-ice habitat constitutes an important shelter and nursery ground for Antarctic krill (David et al., 2021, Meyer et al., 2017). At the same time, sea-ice, its associated communities, related biogeochemical processes and trophic interactions are highly sensitive to climate-induced changes in structure, temporal dynamics and spatial extent."*

**K. HEYWOOD:** Sea and shelf ice are mentioned as an object of research in the objectives chapter of all three themes. How will you be accessing ice stations? How will you choose locations for sea ice stations? Will they be on fast ice or ice floes? How frequently will they be monitored? Throughout the year? More explanation is needed (probably in section 3).

***REPLY***: *A paragraph has been extended on the coring of sea-ice in 3.1.3 to answer most of the questions above also referring to biological studies. "Open ocean observations of cryospheric components will be complemented by sea-ice coring for direct physical, biological, and chemical material collection and direct data measurements should be conducted in all three sub-areas, primarily in the Atka Bay and preferably at the same or similar locations and at the same time of year on a regular basis from the ship, with helicopter support if necessary. Sampling shall be done on different ice types, including fast ice, seasonal ice, snow cover and platelet ice, where these exist and sub-sea ice ocean properties from manual CTD casts Arndt et al., 2020)." A cross reference is made in 3.2.2, where another paragraph has been added in more detail sampling and observation technology below the sea-ice (coring technology had already been mentioned), see also above: "Autonomous bio-environmental observatories shall constitute an important pillar of LTER at the IEAMO. By combining multiple sensors on bottom-moored, sea-ice-moored and free drifting platforms, the spatio-temporal gaps between field campaigns will be covered with high-resolution data. These systems should merge existing state-of-the-art environmental sensors, such as CTDs, nutrient sensors, fluorometers, spectral radiometers and optical sediment traps, with the newest technology to monitor higher organisms beyond microbes. Examples for such sensors are camera systems, autonomous multi-frequency echosounders (e.g. Acoustic Zooplankton and Fish Profilers, Wideband Autonomous Transceiver) able to record and transmit data and to receive real-time manipulation of the sampling programme, automatic eDNA samplers and imaging profilers (e.g. Underwater Vision Profiler)."*

**K. HEYWOOD:** Section 4.3 would benefit from references.

***REPLY:*** *A few overarching references are added.*

**Minor comments:**

**K. HEYWOOD:** L34 I wouldn't call it easterly – not clear what is meant here – omit?

***REPLY:*** *For a better understanding changed to "to the east". To be precise this cannot be omitted because an adjacent region could also lay in the West of the eastern Weddell Sea.*

**K. HEYWOOD:** L37 I didn't understand what was meant by "and sound data can act as a model to develop and calibrate projections". What do you mean by sound? How can data act as anything? How can a model develop or calibrate a projection?

*REPLY: Text improved: "...consistent and comparable time series data underpinning and testing…"*

**K. HEYWOOD:** L108-110 Repetition of HAFOS, rephrase sentence.

*REPLY: Text revised accordingly*

**K. HEYWOOD:** L125 can you give an idea of how many "a number" will be?

*REPLY: We included a reference to Fig. 2(b) because there are different types of sites/stations, such as core, representative and optional stations and, thus, no concrete numbers can be mentioned here without going into too much detail and repeat, what is shown in Fig. 2b.*

**K. HEYWOOD:** L126 what does sensu mean?

*REPLY: Changed to "as described by"*

**K. HEYWOOD:** L182- 183 You have east and west round the wrong way – the ASC flows west and the ACC flows east.

*REPLY: Corrected accordingly*

**K. HEYWOOD:** Numbering of section 3.3.2 is incorrect, p23,´

*REPLY: Corrected*

**K. HEYWOOD:** L710 I wasn't sure what this sentence meant. Grammatically the first line (l710) didn't seem to make sense.

*REPLY: Changed to: "In the first Antarctic and Southern Ocean Horizon Scan of SCAR (Kennicutt et al., 2014), the scientific community identified the need for a better understanding of systems, which can only be achieved by targeted long-term observations, measurements and analyses."*

**Figures:**

**K. HEYWOOD:** Fig.1 This is a nice introductory figure. However it is not much described. To be useful to readers, especially those who do not know the currents and water masses, this figure needs further explanation. The relevance of the water masses and circulation to the ecology/biogeochemistry should be stated.

*REPLY: We cannot repeat all information from the main text in section 3.1.1 in the figure caption.  However, we added some text on the water flow along the coast through the IEAMO: "Possible location of the IEAMO region within the East Antarctic Southern Ocean. Arrows indicate large-scale advective water-mass pathways. Deep water entering the*

*Weddell gyre from the ACC joins the southern limb of the gyre, of which the ACoC is also a part. After leaving the IEAMO region, the water flow continues along the slope and shelves. Interaction with the broad shelves in the south leads to ISW, a predecessor of Antarctic Bottom Water. Small green circles indicate sites of ongoing mooring programs."*

**K: HEYWOOD:** Fig. 2. Label for Kapp Norwegia is needed. The caption is confusing because it says there are three areas but the highlighted region is all one? Are the three labels meant to be on top of the survey sites? I'd like to see a map where the boundaries and properties for the three regions are clearly shown. It might be helpful to have a map of the three regions showing sea ice minima and maxima, SST, or something similar?

**REPLY:** *The figure was revised and shows now more details (see above) and more clearly the three areas of investigation. The revised caption starts with "(a) Geographic position of the components of the IEAMO. The three circles represent three possible sub-areas, off the Fimbul and Ekström ice shelves, respectively, as well as off Kapp Norvegia. The highlighted area shows the wider IEAMO region, where large-scale data from methods like remote sensing and bathymetry are important for most other specific measurements."*

*We cannot superimpose the existing bathymetry by SST or sea-ice cover, since then the figure would become too complex and confusing. We included a figure (new Fig. 3) displaying some of the most important environmental drivers (sea-ice, SST and Chl-a) and a sentence in the text: "For examples of some most important environmental drivers of the marine ecosystem in the area under consideration see Figs. 1 (currents), and Fig. 2. (bathymetry), and Fig. 4 (sea ice, sea surface temperature, and chlorophyll-a)"*

[Figure]

**Figure 3:** Sea ice concentration for the minimum and maximum sea ice extent in 2021, and the summer climatology, i.e. December 21 to March 20, for the time period 2002-2019 for the sea surface chlorophyll a concentration and temperature in the vicinity of the presented IEAMO. Sea ice concentration data were obtained from EUMETSAT Ocean and Sea Ice Satellite Application Facility (OSI SAF, Lavergne et al., 2019). Sea surface temperature and chlorophyll a concentration were obtained from the ocean color data distribution site: http://oceandata.sci.gsfc.nasa.gov/.

**K. HEYWOOD: Fig 3 -** check that the acronym soup is already defined, and define in caption for maximum readability.

***REPLY:*** *All acronyms are explained in the revised caption.*

**K. HEYWOOD:** I couldn't find where figure 4 is discussed?

***REPLY:*** *A sentence is included: "To ensure synergistic effects for science and marine conservation policy, the location of the IEAMO is partly congruent with the proposed WSMPA (Fig. 5)." Please note, the figure had not been criticized but we adapted it to the style of a true scientific paper and are convinced that this figure improved considerably.*

Karen J. Heywood

 University of East Anglia